# Putting perception into action with inverse optimal control for continuous psychophysics

**Dominik Straub[1,2]\*, Constantin A Rothkopf[1,2,3]\***

[1]Centre for Cognitive Science, Technical University of Darmstadt, Darmstadt, Germany; [2]Institute of Psychology, Technical University of Darmstadt, Darmstadt, Germany; [3]Frankfurt Institute for Advanced Studies, Goethe University Frankfurt, Frankfurt, Germany

**\*For correspondence:**
dominik.straub@tu-darmstadt.de (DS);
rothkopf@psychologie.tu-darmstadt.de (CAR)

**Competing interest:** The authors declare that no competing interests exist.

**Abstract** Psychophysical methods are a cornerstone of psychology, cognitive science, and neuroscience where they have been used to quantify behavior and its neural correlates for a vast range of mental phenomena. Their power derives from the combination of controlled experiments and rigorous analysis through signal detection theory. Unfortunately, they require many tedious trials and preferably highly trained participants. A recently developed approach, continuous psychophysics, promises to transform the field by abandoning the rigid trial structure involving binary responses and replacing it with continuous behavioral adjustments to dynamic stimuli. However, what has precluded wide adoption of this approach is that current analysis methods do not account for the additional variability introduced by the motor component of the task and therefore recover perceptual thresholds that are larger compared to equivalent traditional psychophysical experiments. Here, we introduce a computational analysis framework for continuous psychophysics based on Bayesian inverse optimal control. We show via simulations and previously published data that this not only recovers the perceptual thresholds but additionally estimates subjects' action variability, internal behavioral costs, and subjective beliefs about the experimental stimulus dynamics. Taken together, we provide further evidence for the importance of including acting uncertainties, subjective beliefs, and, crucially, the intrinsic costs of behavior, even in experiments seemingly only investigating perception.

## Editor's evaluation

This important article presents a Bayesian model framework for estimating individual perceptual uncertainty from continuous tracking data – a powerful and exciting alternative to traditional binary-choice psychophysical experiments. The model takes into account motor variability, action cost, and possible misestimation of the generative dynamics. The analyses provide compelling evidence for the framework. The article is clearly written and provides a didactic resource for students wishing to implement similar models on continuous action data.

## Introduction

Psychophysical methods such as forced-choice tasks are widely used in psychology, cognitive science, neuroscience, and behavioral economics because they provide precise and reliable quantitative measurements of the relationship between external stimuli and internal sensations (*Gescheider, 1997*; *Wichmann and Jäkel, 2018*). The tasks employed by traditional psychophysics are typically characterized by a succession of hundreds of trials in which stimuli are presented briefly and the

**eLife digest** Humans often perceive the world around them subjectively. Factors like light brightness, the speed of a moving object, or an individual's interpretation of facial expressions may influence perception. Understanding how humans perceive the world can provide valuable insights into neuroscience, psychology, and even people's spending habits, making human perception studies important. However, these so-called psychophysical studies often consist of thousands of simple yes or no questions, which are tedious for adult volunteers, and nearly impossible for children.

A new approach called 'continuous psychophysics' makes perception studies shorter, easier, and more fun for participants. Instead of answering yes or no questions (like in classical psychophysics experiments), the participants follow an object on a screen with their fingers or eyes. One question about this new approach is whether it accounts for differences that affect how well participants follow the object. For example, some people may have jittery hands, while others may be unmotivated to complete the task.

To overcome this issue, Straub and Rothkopf have developed a mathematical model that can correct for differences between participants in the variability of their actions, their internal costs of actions, and their subjective beliefs about how the target moves. Accounting for these factors in a model can lead to more reliable study results. Straub and Rothkopf used data from three previous continuous psychophysics studies to construct a mathematical model that could best predict the experimental results. To test their model, they then used it on data from a continuous psychophysics study conducted alongside a classical psychophysics study. The model was able to correct the results of the continuous psychophysics study so they were more consistent with the results of the classical study.

This new technique may enable wider use of continuous psychophysics to study a range of human behavior. It will allow larger, more complex studies that would not have been possible with conventional approaches, as well as enable research on perception in infants and children. Brain scientists may also use this technique to understand how brain activity relates to perception.

subject responds with a binary decision, for example, whether target stimuli at different contrast levels were perceived to be present or absent.

The analysis of such experimental data using signal detection theory (SDT; *Green and Swets, 1966*) employs a model that assumes that a sensory signal generates an internal representation corrupted by Gaussian variability. A putative comparison of this stochastic signal with an internal criterion leads to a binary decision. These assumptions render human response rates amenable to analysis based on Bayesian decision theory, by probabilistically inverting the model of the decision-generating process. This provides psychologically interpretable measures of perceptual uncertainty and of the decision criterion, for example, for contrast detection. Thus, the power of classical psychophysics derives from the combination of controlled experimental paradigms with computational analysis encapsulated in SDT.

Psychophysical experiments and their analysis with SDT have revealed invaluable knowledge about perceptual and cognitive abilities and their neuronal underpinnings, and found widespread application to tasks as diverse as eyewitness identification and medical diagnosis (for recent reviews, see, e.g., *Lynn and Barrett, 2014*; *Wixted, 2020*). One central drawback, however, is that collecting data in such tasks is often tedious, as famously noted already by *James, 1890*. This leads to participants' engagement levels being low, particularly in untrained subjects, resulting in measurements contaminated by additional variability (*Manning et al., 2018*). A common solution is to rely on few but highly trained observers to achieve consistent measurements (*Green and Swets, 1966*; *Jäkel and Wichmann, 2006*).

Recent work has suggested overcoming this shortcoming by abandoning the rigid structure imposed by independent trials and instead eliciting continuous behavioral adjustments to dynamic stimuli (*Bonnen et al., 2015*; *Bonnen et al., 2017*; *Knöll et al., 2018*; *Huk et al., 2018*). In their first study using continuous psychophysics, *Bonnen et al., 2015*, used a tracking task, in which subjects moved a computer mouse to track targets of different contrasts moving according to a random walk governed by linear dynamics with additive Gaussian noise. This experimental protocol not only

requires orders of magnitude less time compared to traditional psychophysical methods, but participants report it to be more natural, making it more suitable for untrained subjects and potentially children and infants. Similar approaches have recently been used to measure contrast sensitivity (*Mooney et al., 2018*), eye movements toward optic flow (*Chow et al., 2021*), or retinal sensitivity (*Grillini et al., 2021*).

To obtain measures of perceptual uncertainty from the tracking data, *Bonnen et al., 2015*, proposed analyzing these data with the Kalman filter (KF; *Kalman, 1960*), the Bayes-optimal estimator for sequential observations with Gaussian uncertainty. The KF, however, models the perceptual side of the tracking task only, that is, how an ideal observer (*Geisler, 1989*) sequentially computes an estimate of the target's position. Unfortunately, although the perceptual thresholds estimated with the KF from the tracking task were highly correlated with the perceptual uncertainties obtained with an equivalent classical forced-choice task employing stimuli with the same contrast, *Bonnen et al., 2015*, reported that these were systematically larger by a large margin.

This is not surprising since every perceptual laboratory task involves some motor control component at least for generating a response, albeit to different degrees. While a detection task may require pressing a button, a reproduction task involves a motor control component, and a tracking task involves continuous visuomotor control, as mentioned by *Bonnen et al., 2015*. In addition to the problem of estimating the current position of the target, a tracking task encompasses the motor control problem of moving the finger, computer mouse, or gaze toward the target. This introduces additional sources of variability and bias. First, repeated movements toward a target exhibit variability (*Faisal et al., 2008*), which arises because of neural variability during execution of movements (*Jones et al., 2002*) or their preparation (*Churchland et al., 2006*). Second, a subject might trade off the instructed behavioral goal of the tracking experiment with subjective costs, such as biomechanical energy expenditure (*di Prampero, 1981*) or mental effort (*Shenhav et al., 2017*). Third, subjects might have mistaken assumptions about the statistics of the task (*Petzschner and Glasauer, 2011*; *Beck et al., 2012*), which can lead to different behavior from a model that perfectly knows the task structure, that is, ideal observers. A model that only considers the perceptual side of the task will, therefore, tend to overestimate perceptual uncertainty because these additional factors get lumped into perceptual model parameters, as we will show in simulations.

Here, we account for these factors by applying ideas from optimal control under uncertainty (see, e.g., *Wolpert and Ghahramani, 2000*; *Todorov and Jordan, 2002*; *Shadmehr and Mussa-Ivaldi, 2012*, for reviews in the context of sensorimotor neuroscience) to the computational analysis of continuous psychophysics. Partially observable Markov decision processes (POMDPs; *Åström, 1965*; *Kaelbling et al., 1998*) offer a general framework for modeling sequential perception, and actions under sensory uncertainty, action variability, and explicitly include behavioral costs (*Hoppe and Rothkopf, 2019*). They can be seen as a generalization of Bayesian decision theory, on which classical analysis of psychophysics including SDT is based, to sequential tasks. Specifically, we employ the well-studied linear quadratic Gaussian framework (LQG; *Anderson and Moore, 2007*), which accommodates continuous states and actions under linear dynamics and Gaussian variability. While the KF, being the Bayes-optimal estimator for a linear-Gaussian system, can be shown to minimize the squared error between the latent target and its estimate, the optimal control framework can flexibly incorporate different kinds of costs such as the intrinsic costs of actions. Thus, it can accommodate task goals beyond estimation as well as physiological and cognitive costs of performing actions.

Modeling the particular task as it is implemented by the researcher allows deriving a normative model of behavior such as ideal observers in perceptual science (*Green and Swets, 1966*; *Geisler, 1989*) or optimal feedback control models in motor control (*Wolpert and Ghahramani, 2000*; *Todorov and Jordan, 2002*). This classic task analysis at the computational level (*Marr, 1982*) can now be used to produce predictions of behavior, which can be compared to the actual empirically observed behavior. However, the fundamental assumption in such a setting is that the subject is carrying out the instructed task and that subject's internal model of the task is identical to the underlying generative model of the task employed by the researcher. Here, instead, we allow for the possibility that the subject is not acting on the basis of the model the researcher has implemented in the experiment. Instead, we allow for the possibility that subjects' cost function does not only capture the instructed task goals but also the experienced subjective cost such as physiological and cognitive costs of performing actions. Such an approach is particularly useful in a more naturalistic task setting,

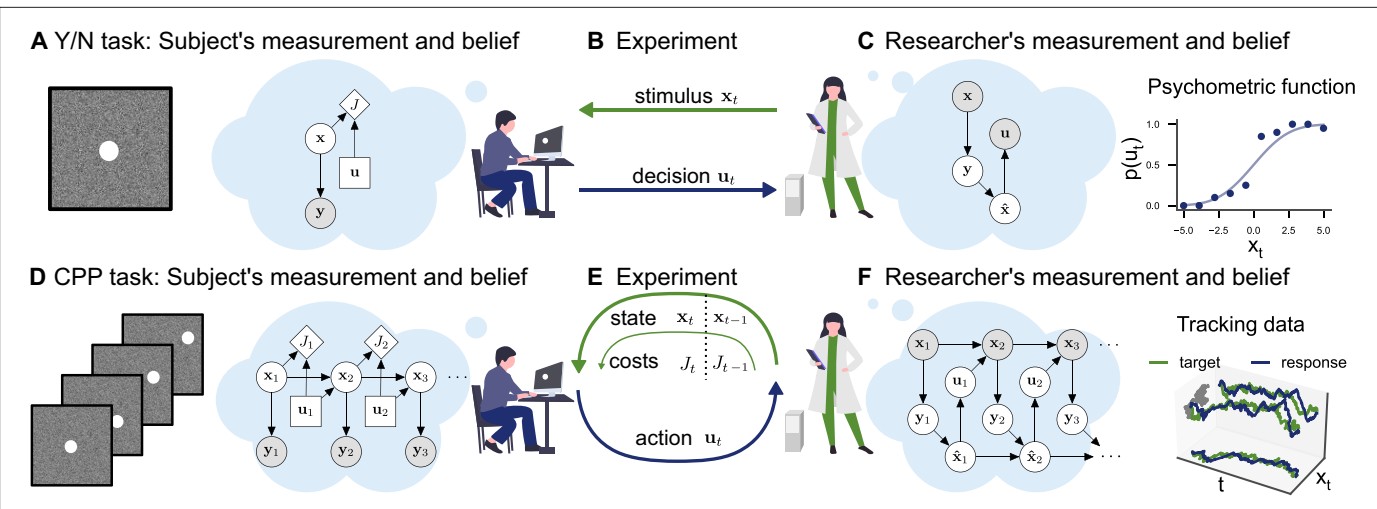

**Figure 1.** Conceptual frameworks for classical and continuous psychophysics. (**A**) In a classical psychophysics task, the subject receives stimuli $x_t$ on independent trials, generates sensory observations $y_t$, forms beliefs about the stimulus $\hat{x}_t$, and (**B**) makes a single decision $u_t$ (e.g., whether the target stimulus was present or absent). (**C**) The researcher has a model of how the agent makes decisions and measures the subject's sensitivity and decision criterion by inverting this model. For example, one can estimate the agent's visual uncertainty by computing the width of a psychometric function. (**D**) In a continuous psychophysics task, a continuous stream of stimuli is presented. The subject has an internal model of the dynamics of the task, which they use to form a belief about the state of the world and then perform continuously actions (**E**) based on their belief and subjective costs. (**F**) The researcher observes the subject's behavior and inverts this internal model, fo example, using Bayesian inference applied to optimal control under uncertainty.

where subjective costs and benefits of behavior are difficult to come by a priori. Similarly, we allow subjects' subjective internal model of stimulus dynamics to differ from the true model employed by the researcher. In the spirit of rational analysis (*Simon, 1955*; *Anderson, 1991*; *Gershman et al., 2015*), we subsequently invert this model of human behavior by developing a method for performing Bayesian inference of parameters describing the subject. Importantly, inversion of the model allows all parameters to be inferred from behavioral data and does not presuppose their value, so that, for example, if subjects' actions were not influenced by subjective internal behavioral costs, the internal cost parameter would be estimated to be zero. This approach therefore reconciles normative and descriptive approaches to sensorimotor behavior as it answers the question which subjective internal model, beliefs, and costs best explain observed behavior.

We show through simulations with synthetic data that models not accounting for behavioral cost and action variability overestimate perceptual uncertainty when applied to data that includes these factors. We apply our method to data from three previously published experiments, infer perceptual uncertainty, and obtain overwhelming evidence through Bayesian model comparison with the widely applicable information criterion (WAIC; *Watanabe and Opper, 2010*) that this model explains the data better than models only considering perceptual contributions to behavioral data. Additionally, the method provides inferences of subjects' action variability, subjective behavioral costs, and subjective beliefs about stimulus dynamics. Taken together, the methodology presented here improves current analyses and should be the preferred analysis technique for continuous psychophysics paradigms.

## Results
### Computational models of psychophysical tasks

In a classical psychophysics task (e.g., position discrimination; *Figure 1A and B*), the stimuli $x_t$ presented to the observer are usually independent and identically distributed between trials. This allows for straightforward application of Bayesian decision theory: in any trial, the observer receives a stochastic measurement $y_t \sim p(y_t|x_t, \boldsymbol{\theta})$, forms a belief $p(x_t|y_t, \boldsymbol{\theta}) \propto p(y_t|x_t, \boldsymbol{\theta})p(x_t, \boldsymbol{\theta})$, and makes a single decision minimizing a cost function $u_t = \arg\min \int J(u_t, x_t, \boldsymbol{\theta})p(x_t|y_t, \boldsymbol{\theta})dx_t$. In SDT, for example, observations are commonly assumed to be Gaussian distributed and the cost function $J$ assigns a value to correct and wrong decisions. These assumptions make it straightforward to compute the observer's decision probabilities $p(u_t|x_t)$ given parameters $\boldsymbol{\theta}$ (e.g., sensitivity and criterion in SDT) and invert the model to

infer those parameters from behavior, yielding a posterior distribution $p(\theta | u_{1:T}, x_{1:T})$ (see **Figure 1C**). Similarly, a psychometric curve can be interpreted as the decision probabilities of a Bayesian observer with a particular choice of measurement distribution, which determines the shape of the curve. The assumption of independence between trials is critical in the application of these modeling techniques.

Continuous psychophysics abandons the independence of stimuli between individual trials. Instead, stimuli are presented in a continuous succession. For example, in a position-tracking task, the stimulus moves from frame to frame and the observer's goal is to track the stimulus with their mouse cursor (**Figure 1D and E**). In a computational model of a continuous task, the stimulus dynamics are characterized by a state transition probability $p(\mathbf{x}_t | \mathbf{x}_{t-1}, \mathbf{u}_{t-1}, \theta)$ and can potentially be influenced by the agent's actions $\mathbf{u}_{t-1}$. As in the trial-based task, the agent receives a stochastic measurement $\mathbf{y}_t \sim p(\mathbf{y}_t | \mathbf{x}_t, \theta)$ and has the goal to perform a sequence of actions $\mathbf{u}_{1:T} = \arg\min \int J(\mathbf{u}_{1:T}, \mathbf{x}_{1:T}) p(\mathbf{x}_{1:T} | \mathbf{y}_{1:T}) d\mathbf{x}_{1:T}$, minimizing the cost function. Formally, this problem of acting under uncertainty is a POMDP and is computationally intractable in the general case (**Åström, 1965**; **Kaelbling et al., 1998**).

In target-tracking tasks used in previous continuous psychophysics studies (**Bonnen et al., 2015**; **Bonnen et al., 2017**; **Huk et al., 2018**; **Knöll et al., 2018**), in which the target is on a random walk, the dynamics of the stimulus are linear and the subjects' goal is to track the target. Tracking can be reasonably modeled with a quadratic cost function penalizing the separation between the target and the position of the tracking device. The variability in the dynamics and the uncertainty in the observation model can be modeled with Gaussian distributions. The resulting LQG control problem is a well-studied special case of the general POMDP setting, which can be solved exactly by determining an optimal estimator and an optimal controller (**Edison, 1971**; **Anderson and Moore, 2007**). It is defined by a discrete-time linear dynamical system with Gaussian noise,

$$\mathbf{x}_t = A\mathbf{x}_{t-1} + B\mathbf{u}_{t-1} + V\epsilon_t, \ \epsilon_t \sim \mathcal{N}(0, I), \tag{1}$$

a linear observation model with Gaussian noise,

$$\mathbf{y}_t = C\mathbf{x}_t + W\eta_t, \ \eta_t \sim \mathcal{N}(0, I), \tag{2}$$

and a quadratic cost function,

$$J(\mathbf{u}_{1:T}) = \sum_{t=1}^{T} \left[ \mathbf{x}_t^T Q \mathbf{x}_t + \mathbf{u}_t^T R \mathbf{u}_t \right], \tag{3}$$

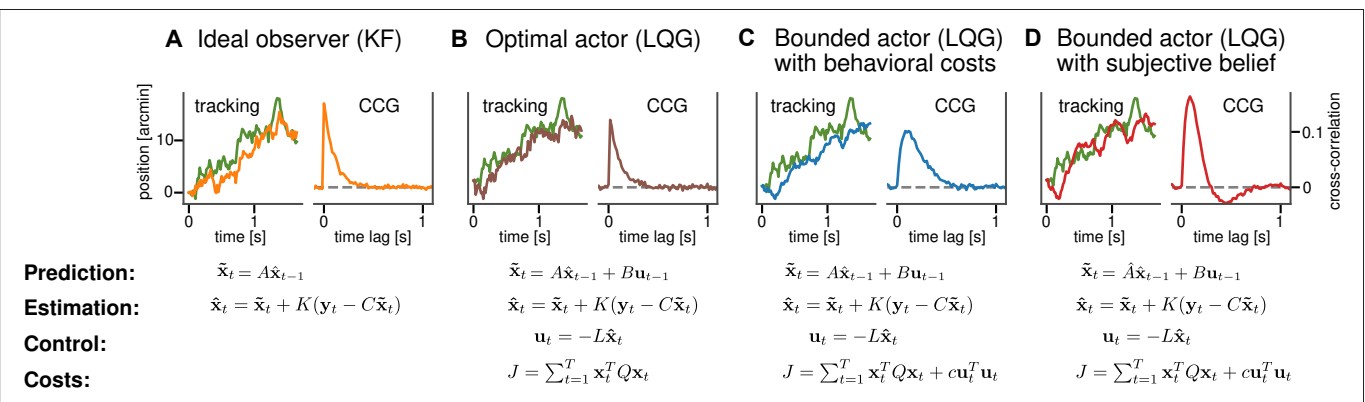

| | **A** Ideal observer (KF) | **B** Optimal actor (LQG) | **C** Bounded actor (LQG) with behavioral costs | **D** Bounded actor (LQG) with subjective belief |
|---|---|---|---|---|
| **Prediction:** | $\tilde{\mathbf{x}}_t = A\hat{\mathbf{x}}_{t-1}$ | $\tilde{\mathbf{x}}_t = A\hat{\mathbf{x}}_{t-1} + B\mathbf{u}_{t-1}$ | $\tilde{\mathbf{x}}_t = A\hat{\mathbf{x}}_{t-1} + B\mathbf{u}_{t-1}$ | $\tilde{\mathbf{x}}_t = \hat{A}\hat{\mathbf{x}}_{t-1} + B\mathbf{u}_{t-1}$ |
| **Estimation:** | $\hat{\mathbf{x}}_t = \tilde{\mathbf{x}}_t + K(\mathbf{y}_t - C\tilde{\mathbf{x}}_t)$ | $\hat{\mathbf{x}}_t = \tilde{\mathbf{x}}_t + K(\mathbf{y}_t - C\tilde{\mathbf{x}}_t)$ | $\hat{\mathbf{x}}_t = \tilde{\mathbf{x}}_t + K(\mathbf{y}_t - C\tilde{\mathbf{x}}_t)$ | $\hat{\mathbf{x}}_t = \tilde{\mathbf{x}}_t + K(\mathbf{y}_t - C\tilde{\mathbf{x}}_t)$ |
| **Control:** | | $\mathbf{u}_t = -L\hat{\mathbf{x}}_t$ | $\mathbf{u}_t = -L\hat{\mathbf{x}}_t$ | $\mathbf{u}_t = -L\hat{\mathbf{x}}_t$ |
| **Costs:** | | $J = \sum_{t=1}^{T} \mathbf{x}_t^T Q \mathbf{x}_t$ | $J = \sum_{t=1}^{T} \mathbf{x}_t^T Q \mathbf{x}_t + c\mathbf{u}_t^T \mathbf{u}_t$ | $J = \sum_{t=1}^{T} \mathbf{x}_t^T Q \mathbf{x}_t + c\mathbf{u}_t^T \mathbf{u}_t$ |

**Figure 2.** Computational models for continuous psychophysics. (**A**) In the Kalman filter (KF) model, the subject makes an observation $\mathbf{y}_t$ with Gaussian variability at each time step. They combine their prediction $\tilde{\mathbf{x}}_t$ with their observation to compute an optimal estimate $\hat{\mathbf{x}}_t$. (**B**) In the optimal control models, this estimate is then used to compute an optimal action $\mathbf{u}_t$ using the linear quadratic regulator (LQR). The optimal action can be based on the task goal only or (**C**) bounded by internal costs, which, for example, penalize large movements. (**D**) Finally, the subject may act rationally using optimal estimation and control, but may use a subjective internal model of stimulus dynamics that differs from the true generative model of the task. These four different models are illustrated with an example stimulus and tracking trajectory (left subplots) and corresponding cross-correlograms (CCG, right subplots; see **Mulligan et al., 2013** and 'Cross-correlograms').

The online version of this article includes the following figure supplement(s) for figure 2:

**Figure supplement 1.** Pareto efficiency plot.

where $Q$ defines the costs for the state (e.g., squared distance between the target and a mouse or hand) and $R$ defines the cost of actions (biomechanical, cognitive, etc.).

For the LQG control problem, the *separation principle* between estimation and control holds (**Davis and Vinter, 1985**). This means that the optimal solutions for the estimator (KF) and controller (LQR) can be computed independently from one another. The KF (**Kalman, 1960**) is used to form a belief $\hat{\mathbf{x}}_t$ about the current state based on the previous estimate and the current observation:

$$\hat{\mathbf{x}}_t = A\hat{\mathbf{x}}_{t-1} + B\mathbf{u}_{t-1} + K\left(\mathbf{y}_t - C\left(A\hat{\mathbf{x}}_{t-1} + B\mathbf{u}_{t-1}\right)\right). \tag{4}$$

This belief is then used to compute the optimal action $\mathbf{u}_t$ based on the linear quadratic regulator (LQR, **Kalman, 1964**): $\mathbf{u}_t = -L\hat{\mathbf{x}}_t$. For the derivations of $K$ and $L$, see section 'LQG control derivations'.

We consider four models, each of which is a generalization of the previous one. Importantly, this means that the most general model contains the remaining three as special cases. We introduce the models in the following paragraphs and describe their respective free parameters. For a complete specification of how these parameters need to be set to model a tracking task and how this gives rise to the matrices defining the linear dynamical systems for each model, see *Appendix 2—table 1*.

First, the ideal observer model in the tracking task is the KF as it is the optimal Bayesian sequential estimator of the latent, unobserved state of the system (*Figure 2A*). The state represents the current position of the target and there is no explicit representation of the agent's response or action. The computational goal of the KF is to perform sequential inference about the position of the target given sensory measurements and an internal model of the target's dynamics. As a proxy for a behavioral response, one can therefore use the best estimate $\hat{\mathbf{x}}_t$ that corresponds to the target's position. Because the ideal observer assumption implies that the task dynamics, that is, the standard deviation of the random walk are perfectly known to the subject, the KF has only one free parameter, the perceptual uncertainty $\sigma$. This parameter describes the uncertainty in the sensory observation of target's true position.

Second, the optimal actor model maintains a representation of its own response, that is, the position of the mouse cursor in addition to the target's position: the state space is now two-dimensional. This allows us, in addition to the sensory uncertainty about the target, to model sensory uncertainty about the position of the cursor $\sigma_p$ (*Figure 2B*). It is now possible to explicitly model a behavioral goal different from optimal sensory inference, as in the case of the KF model. The behavioral goal is encapsulated in the cost function and for the optimal actor the cost function is fully determined by the task goal intended by the researcher, that is, the tracking of the target. Therefore, the cost function does not contain any free parameters in the optimal actor model. Different from the KF model that only carries out estimation, the optimal actor model generates a sequences of actions with their associated action variability $\sigma_m$. The optimal behavior of the model is now to move the cursor as closely as possible toward the estimated position of the target.

Third, the bounded actor model is based on the optimal actor model by also using a state representation involving the target and its own response as well as uncertain sensory observations of both these state variables. We use the term "bounded actor" as it is customary in parts of the cognitive science literature, where, according to Herbert Simon's definition (**Simon, 1955**), bounded rationality, which is related to the concept of resource-rationality, takes the limitations on cognitive capacity into a account. It is also a control model so that sequential actions are generated. The difference lies in the cost function, which now can additionally accommodate internal behavioral costs $c$ (*Figure 2C*) that may penalize, for example, large actions and thus result in a larger lag between the response and the target. The control cost parameter $c$ therefore can implement a trade-off between tracking the target and expending effort (see *Figure 2—figure supplement 1*).

Finally, the subjective actor is a version of the bounded actor but now bases its decisions on an internal model of the stimulus dynamics that may differ from the true generative model employed by the experimenter. The causes for differences in the internal model may be sensory, perceptual, cognitive, or stemming from constraints on planning. For example, in an experimental design with a target moving according to a random walk on position (**Bonnen et al., 2015**) with a true standard deviation $\sigma_{\mathrm{rw}}$, the subject could instead assume that the target follows a random walk on position with standard deviation $\sigma_s$ and an additional random walk on the velocity with standard deviation $\sigma_v$. These subjective assumptions about the target dynamics can, for example, lead the subjective actor to overshoot the target, for which they then have to correct (*Figure 2D*). For the subjective actor, the

state space is three-dimensional because it contains the target's velocity in addition to the target and response positions (see Appendix 2). One could also formulate a version of any of the other models to include subjective beliefs. However, the subjective actor model is a generalization that includes all of these possible models. Accordingly, if the subjective beliefs played an important role but the costs or variability did not, we would infer the relevant parameters to have a value close to zero.

## Bayesian inverse optimal control

The normative models described in the previous section treat the problem from the subject's point of view. That is, they describe how optimal actors with different internal models and different goals as encapsulated by the cost function should behave in a continuous psychophysics task. The above normative models may give rise to different sequences of actions, which are the optimal solutions given the respective models with their associated parameters and computational goals. From the subject's perspective, the true state of the experiment $\mathbf{x}_t$ is only indirectly observed via the uncertain sensory observation $\mathbf{y}_t$ (see *Figure 1D*). From the point of view of a researcher, the true state of the experiment is observed because it is under control of the experiment, for example, using a computer that presents the target and mouse position. The computational goal for the researcher is to estimate parameters $\theta$ that describe the perceptual, cognitive, and motor processes of the subject for each model, given observed trajectories $\mathbf{x}_{1:T}$ (*Figure 1F*). In the case of continuous psychophysics, these parameters include the perceptual uncertainty about the target $\sigma$ and about one's own cursor position $\sigma_p$, the control cost $c$, and the action variability $\sigma_m$, as well as parameters that describe how the subject's internal model differs from the true generative model of the experiment employed by the researcher. These properties are not directly observed and can only be inferred from the subject's behavior. Importantly, in the framework presented here, we subsequently need to carry out model comparison across the considered models potentially describing the subject as we do not assume to know the model describing the subject's internal beliefs, costs, and model a priori.

To compute the posterior distribution according to Bayes' theorem $p(\theta|\mathbf{x}_{1:T}) \propto p(\mathbf{x}_{1:T}|\theta)p(\theta)$, we need the likelihood $p(\mathbf{x}_{1:T}|\theta)$, which we can derive using the probabilistic graphical model from the researcher's point of view (*Figure 1F*). The graph's structure is based on that describing the subject's point of view. However, because different variables are observed, the decoupling of the perceptual and control processes no longer holds and we need to marginalize out the latent internal observations of the subject: $p(\mathbf{x}_{1:T}|\theta) = \int (\mathbf{x}_{1:T}, \mathbf{y}_{1:T}|\theta)d\mathbf{y}_{1:T}$. The section 'Likelihood function' describes how to do this efficiently. We compute posterior distributions using probabilistic programming and Bayesian

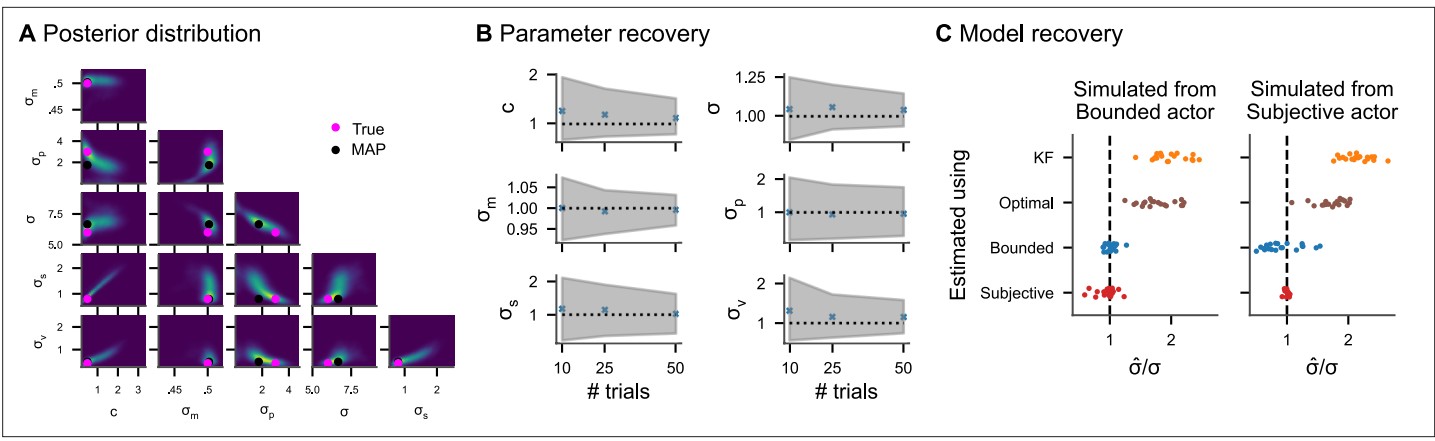

**Figure 3.** Inference on simulated data. (**A**) Pairwise joint posterior distributions (0.5, 0.9, and 0.99 highest density intervals) inferred from simulated data from the subjective actor with parameters representative of real tracking data ($c = 0.5$, $\sigma = 6.0$, $\sigma_p = 3.0$, $\sigma_m = 0.5$, $\sigma_s = 0.8$, and $\sigma_v = 0.4$). The pink dots mark the true value used to simulate the data, while the black dots mark the posterior mode. (**B**) Average posterior means and average 95% credible intervals relative to the true value for different numbers of trials (200 repetitions each). (**C**) Model recovery analysis. Inferring perceptual uncertainty ($\hat{\sigma}/\sigma$, posterior mean relative to the true value) with each of the four models from data simulated from the bounded actor model and subjective actor model, respectively.

The online version of this article includes the following figure supplement(s) for figure 3:

**Figure supplement 1.** Model comparison for simulations.

inference (see section 'Bayesian inference'). In the following, we show empirically through simulations involving synthetic data that the models' parameters can indeed be recovered with Bayesian inverse optimal control.

## Simulation results

To establish that the inference algorithm can recover the model parameters from ground truth data, we generated 200 sets of parameters from uniform distributions resulting in realistic tracking data (see Appendix 3, 'Parameter recovery' for details). For each set of parameters, we simulated datasets consisting of 10, 25, and 50 trials (corresponding to 2, 5, and 10 min) from the subjective actor model. *Figure 3A* shows one example posterior distribution. Average posterior means and credible intervals relative to the true value of the parameter are shown in *Figure 3B*. With only 2 min of tracking, the 95% posterior credible intervals of the perceptual uncertainty $\sigma$ are [0.8, 1.3] relative to the true value on average. This means that the tracking data of an experiment lasting 2 min is sufficient to obtain posterior distributions over model parameters for which 95% of probability is within a range of 20% underestimation and 30% overestimation of the true values. As a comparison, we consider the simulations for estimating the width of a psychometric function in classical psychophysical tasks conducted by *Schütt et al., 2015*. With 800 forced-choice trials (corresponding to roughly 20 min), the 95% posterior credible intervals are within [0.6, 1.6] relative to the true value on average. This suggests a temporal efficiency gain for continuous psychophysics of at least a factor of 10. These simulation results are in accordance with the empirical results of *Bonnen et al., 2015*, who had shown that the tracking task yields stable but biased estimates of perceptual uncertainty in under 2 min of data collection.

Behavior in a task with an interplay of perception and action such as a tracking task contains additional sources of variability and bias beyond sensory influences: repeated actions with the same goal are variable and the cost of actions may influence behavior. We model these factors as action variability $\sigma_m$ and control cost $c$, respectively, in the bounded actor model. A model without these factors needs to attribute all the experimentally measured behavioral biases and variability to perceptual factors, even when they are potentially caused by additional cognitive and motor processes. We

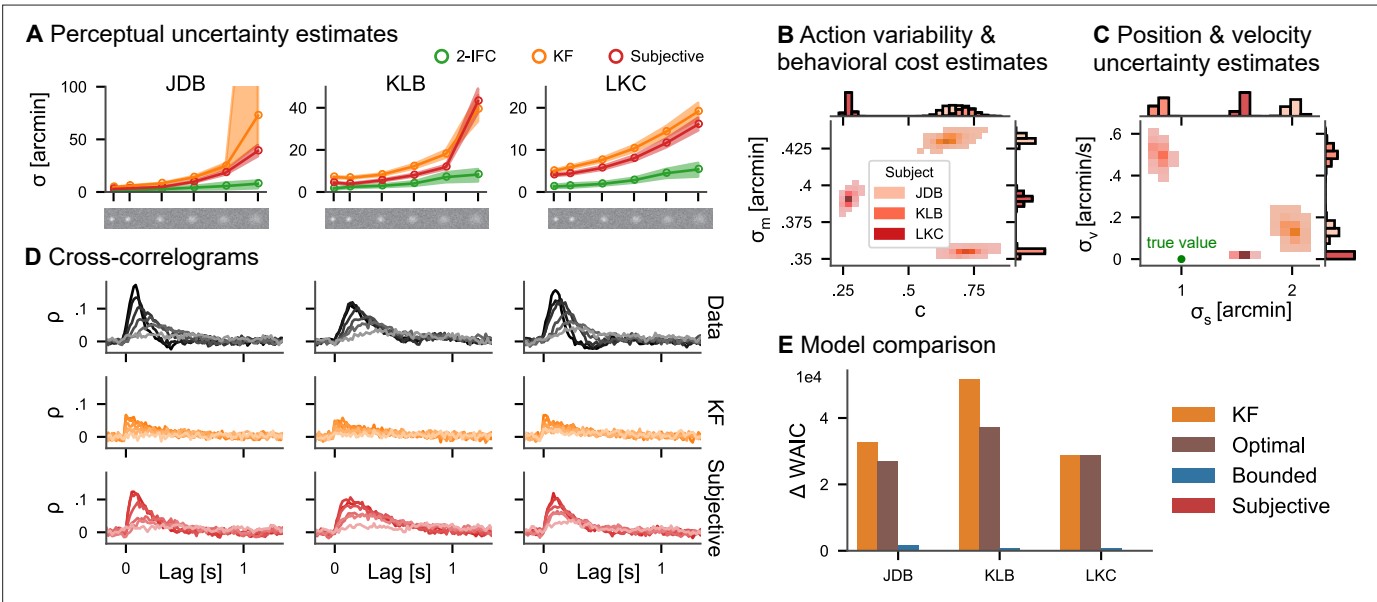

**Figure 4.** Continuous psychophysics. (**A**) Perceptual uncertainty ($\sigma$) parameter estimates (posterior means) for the two-interval forced-choice (2IFC) task and the tracking task (Kalman filter [KF] and linear quadratic Gaussian [LQG] models) in the six blob width conditions (*Bonnen et al., 2015*). The shaded area represents 95% posterior credible intervals. (**B**) Posterior distributions for action cost ($c$) and action variability ($\sigma_m$). (**C**) Posterior distributions for subjective stimulus dynamics parameters (position: $\sigma_s$, velocity: $\sigma_v$). The true values of the target's random walk are marked by a green point. (**D**) Cross-correlograms (CCGs) of the empirical data and both models for all three subjects. (**E**) Model comparison. The difference in widely applicable information criterion (WAIC) w.r.t. the best model is shown with error bars representing WAIC standard error. Models without control cost (KF and optimal) fare worst, while the subjective model has the highest predictive accuracy.

substantiate this theoretical argument via simulations. We simulated 20 datasets with different values for $c$ and $\sigma_m$ sampled from uniform ranges from the bounded actor model and the subjective actor model. Then, we fit each of the four different models presented in *Figure 2* to the tracking data and computed posterior mean perceptual uncertainties. The results are shown in *Figure 3C*. The posterior means of $\sigma$ from the bounded actor and subjective actor scatter around the true value, with the model used to simulate the data being the most accurate for estimating $\sigma$. The other two models (KF and optimal actor), which do not contain intrinsic behavioral costs, both overestimate $\sigma$. This overestimation is worse when the model does also not account for action variability (KF). Furthermore, Bayesian model comparison confirms that the model used to simulate the data is also the model with the highest predictive accuracy in all cases (*Figure 3—figure supplement 1*).

## Continuous psychophysics

We now show that the ability of the optimal control models to account for control costs and action variability results in more accurate estimates of perceptual uncertainty in a target-tracking task that are in closer agreement to the values obtained through an equivalent two-interval forced-choice (2IFC) task. We reanalyze data from three subjects in two previously published experiments (*Bonnen et al., 2015*, Appendix 4, 'Continuous psychophysics' for details): a position discrimination and a position-tracking task. Both experiments employed the same visual stimuli: Gaussian blobs with six different widths within white noise backgrounds were used to manipulate the stimulus contrast (see *Figure 4A*). In the 2IFC discrimination task, the psychophysical measurements of perceptual uncertainty increased with decreasing contrast (*Figure 4A*, green lines). In the tracking experiment, the same visual stimuli moved on a Gaussian random walk with a fixed standard deviation and subjects tracked the target with a small red mouse cursor.

We fit all four models presented in *Figure 2* to the data of the tracking experiment. To a first approximation, it is reasonable to assume that action variability, perceptual uncertainty about the mouse position, and control costs are independent of the target blob width, so that these parameters are shared across the different contrast conditions. The perceptual uncertainty $\sigma$ should be different across contrasts, so we inferred individual parameters $\sigma$ per condition. We assume ideal temporal integration over individual frames, including for the reanalysis of the KF model (see Appendix 4). We focus on the KF and the subjective actor in the following.

The posterior means of estimated perceptual uncertainty in both models were highly correlated with the classical psychophysical measurements ($r > 0.88$ for both models and all subjects). However, the parameters from the subjective actor model are smaller in magnitude than KF's in all three subjects and all but one blob width condition (*Figure 4A*). The reason for this is that the KF attributes all variability to perceptual uncertainty, while the other models explicitly include cognitive and motor influences (see section 'Simulation results'). The average factor between the posterior mean perceptual uncertainty in the continuous task and the 2IFC task in the five higher contrast conditions is 2.39 for the subjective actor, while it is 3.46 for the KF. Only in the lowest contrast condition, it increases to 4.40 and 5.87, respectively. Thus, accounting for action variability and control effort in our model leads to estimates of perceptual uncertainty closer to those obtained with the classical 2IFC psychophysics task. Note that a deviation between the two psychophysical tasks should not be surprising based on previous research comparing the thresholds obtained with different traditional paradigms, as we further discuss below.

Importantly, in addition to the perceptual uncertainty parameter, we also obtain posterior distributions over the other model parameters (*Figure 4B*). The three subjects differ in how variable their actions are ($\sigma_m$) and how much they subjectively penalize control effort ($c$). Additionally, the subjective actor model infers subjects' implicit assumptions about how the target blob moved. The model estimates that two of the three subjects assume a velocity component $\sigma_v$ to the target's dynamics, although the target's true motion according to the experimenter's generative model does not contain such a motion component, while in the third participant this estimated velocity component is close to zero. Similarly, the model also infers the subjectively perceived randomness of the random walk, that is, the standard deviation $\sigma_s$ (*Figure 4C*), for each subject, which can be smaller or larger than the true standard deviation used in generating the stimulus.

As an important test, previous research has proposed comparing the autocorrelation structure of the tracking data with that of the model's prediction (*Bonnen et al., 2015*; *Knöll et al., 2018*;

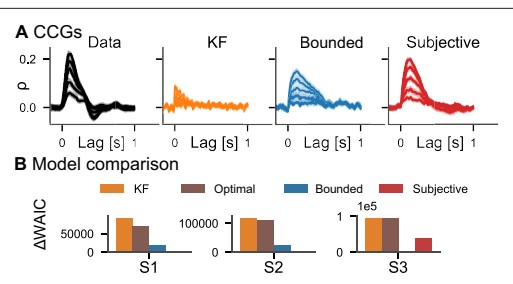

**Figure 5.** Model comparison on motion-tracking data. (**A**) Average cross-correlograms (CCGs) for S1 from **Bonnen et al., 2017** and three models in different target random walk conditions. For the other two subjects, see **Figure 3**. (**B**) Difference in widely applicable information criterion (WAIC) w.r.t. best model.

The online version of this article includes the following figure supplement(s) for figure 5:

**Figure supplement 1.** Cross-correlograms (CCGs) for all subjects.

**Figure supplement 2.** Posterior distributions of model parameters of the subjective model in experiment 2 (**Bonnen et al., 2017**).

**Huk et al., 2018**). We simulated data from both the KF and the subjective actor (20 trials, using the posterior mean parameter estimates) and compared it to the empirical data. To this end, we computed cross-correlograms (CCGs, i.e., the cross-correlation between the velocities of the target and the response, see section 'Cross-correlograms'), for each trial of the real data and simulated data from both models (**Figure 4C**). Indeed, the subjective actor captures more of the autocorrelation structure of the tracking data, quantified by the correlation between the CCGs of the model and those of the data, which was 0.61 for the KF and 0.86 for the subjective actor.

To quantitatively compare the models, we employ Bayesian model comparison using the WAIC (**Watanabe and Opper, 2010**, see section 'Model comparison'). WAIC is computed from posterior samples and estimates the out-of-sample predictive accuracy of a model, thereby taking into account that models with more parameters are more expressive. Overwhelming evidence for the subjective actor model is provided by Akaike weights larger than 0.98 in all three subjects, with the weight for the KF model being equal to 0 within floating point precision. This means that the behavioral data was overwhelmingly more likely under the subjective actor than the KF model, even after taking the wider expressivity of the model into account. Such strong evidence in favor of one of the models is the consequence of the large number of data samples obtained in continuous psychophysics paradigms.

## Application to other datasets

We furthermore applied all four models to data from an additional experiment in which subjects tracked a target with their finger using a motion tracking device. Experiment 2 in the study by **Bonnen et al., 2017** involved five conditions with different standard deviations for the target's random walk. These five conditions were presented in separate blocks (see Appendix 4, 'Motion tracking'). Comparisons of CCGs for one subject are shown in **Figure 5A** (see **Figure 5—figure supplement 1** for all subjects). Qualitatively, the subjective actor captures the cross-correlation structure of the data better than the other models. To quantitatively evaluate the model fits, we again used WAIC (**Figure 5B**). In two of the three subjects, the subjective actor performs best, while in one subject the bounded actor with accurate stimulus model accounts better for the data. We also performed a cross-validation analysis by fitting the models to four of the conditions and evaluating it in the remaining condition. Our reasoning was that the quality of different models in capturing participants' behavior can be quantified by measuring the likelihood of parameters across conditions, that is, by cross-validation, and in the consistency of estimated parameters, that is, their respective variance. The cross-validation analysis provides additional support for the subjective model since it gives more consistent estimates of perceptual uncertainty and has higher log likelihood than the other models (**Appendix 5—figure 1**). Again, we can also inspect and interpret the other model parameters such as control costs, action variability, and subjective belief about target dynamics (**Figure 5—figure supplement 2**). Similarly, when applied to another experiment, in which one human participant, two monkeys, and a marmoset tracked optic flow fields with their gaze (**Knöll et al., 2018**), model selection favored the subjective actor (**Appendix 5—figure 2**).

## Discussion

We introduce an analysis method for continuous psychophysics experiments (*Bonnen et al., 2015*; *Knöll et al., 2018*; *Bonnen et al., 2017*; *Huk et al., 2018*) that allows estimating subjects' perceptual uncertainty, the quantity of interest in the analysis of traditional psychophysics experiments with SDT. We validated the method on synthetic data and demonstrated its feasibility for actions involving human and nonhuman primate participants' tracking with a computer mouse (*Bonnen et al., 2015*), tracking by pointing a finger (*Bonnen et al., 2017*), and tracking using gaze (*Knöll et al., 2018*).

Importantly, we applied our method to experimental data from a manual tracking task, which were collected together with equivalent stimuli in a 2IFC paradigm for the purpose of empirical validation (*Bonnen et al., 2015*). The perceptual uncertainties inferred using inverse optimal control were in better agreement with the 2IFC measurements than those inferred using the previously proposed KF. In the lowest contrast conditions, all models we considered show a large and systematic deviation in the estimated perceptual uncertainty compared to the equivalent 2IFC task. Note that when considering synthetic data we did not see such a discrepancy. Thus, the observed bias points toward additional mechanisms such as a computational cost or computational uncertainty that are not captured by the current models at very low contrast. One reason for this could be that the assumption of constant behavioral costs across different contrast conditions might not hold at very low contrasts because subjects might simply give up tracking the target although they can still perceive its location. Another possible explanation is that the visual system is known to integrate visual signals over longer times at lower contrasts (*Dean and Tolhurst, 1986*; *Bair and Movshon, 2004*), which could affect not only sensitivity in a nonlinear fashion but could also lead to nonlinear control actions extending across a longer time horizon. Further research will be required to isolate the specific reasons. The analysis methods presented in this study are well suited to investigate these questions systematically because they allow modeling these assumptions explicitly and quantifying parameters describing subjects' behavior.

Because subjects' behavior is conceptualized as optimal control under uncertainty (*Åström, 1965*; *Kaelbling et al., 1998*; *Anderson and Moore, 2007*; *Hoppe and Rothkopf, 2019*), the optimal actor model additionally contains action variability and a cost function encapsulating the behavioral goal of tracking the target. The present analysis method probabilistically inverts this model similarly to approaches in inverse reinforcement learning (*Ng and Russell, 2000*; *Ziebart et al., 2008*; *Rothkopf and Dimitrakakis, 2011*) and inverse optimal control (*Chen and Ziebart, 2015*; *Herman et al., 2016*; *Schmitt et al., 2017*). The inferred action variability can be attributed to motor variability (*Faisal et al., 2008*) but also to other cognitive sources, including decision variability (*Gold and Shadlen, 2007*). We extended this model to a bounded actor model in the spirit of rational analysis (*Simon, 1955*; *Anderson, 1991*; *Gershman et al., 2015*) by including subjective costs for carrying out tracking actions. These behavioral costs correspond to intrinsic, subjective effort costs, which may include biomechanical costs (*di Prampero, 1981*), as well as cognitive effort (*Shenhav et al., 2017*), trading off with the subject's behavioral goal of tracking the target.

The model was furthermore extended to allow for the possibility that subjects may act upon beliefs about the dynamics of the target stimuli that differ from the true dynamics employed by the experimenter. Such a situation may arise due to perceptual biases in the observation of the target's dynamics. Model selection using WAIC (*Watanabe and Opper, 2010*) favored this bounded actor with subjective beliefs about stimulus dynamics in all four experiments involving human and monkey subjects with a single exception, in which the bounded actor model better accounted for the behavioral data. Thus, we found overwhelming evidence that subjects' behavior in the continuous psychophysics paradigm was better explained by a model that includes action variability and internal costs for actions increasing with the magnitude of the response.

One possible criticism of continuous psychophysics is that it introduces additional unmeasured factors such as action variability, intrinsic costs, and subjective internal models. While classical psychophysical paradigms take great care to minimize the influence of these factors by careful experimental design, they can nevertheless still be present, for example, as serial dependencies (*Green, 1964*; *Fischer and Whitney, 2014*; *Fründ et al., 2014*). Similarly, estimates of perceptual uncertainty often differ between classical psychophysical tasks when compared directly. For example, *Yeshurun et al., 2008*, reviewed previous psychophysical studies and conducted experiments to empirically test the theoretical predictions of SDT between Yes-No and 2IFC tasks. They found a broad range of deviations

between the empirically measured perceptual uncertainty and the theoretical value predicted by SDT. Similarly, there are differences between 2AFC tasks, in which two stimuli are separated spatially, and their equivalent 2IFC tasks, in which the stimuli are separated in time (*Jäkel and Wichmann, 2006*). While the former task engages spatial attention, the latter engages memory. These factors are typically also not accounted for in SDT-based models. Thus, substantial empirical evidence suggests that factors often not modeled by SDT, such as attention, memory, intrinsic costs, beliefs of the subject differing from the true task statistics, and learning, nevertheless very often influence the behavior in psychophysical experiments. This applies to classical trial-based tasks as well as continuous tasks. Crucially, the framework we employ here allows modeling and inferring some of these additional cognitive factors responsible for some of the observed deviations.

On a conceptual level, the results of this study underscore the fact that psychophysical methods, together with their analysis methods, always imply a generative model of behavior (*Green and Swets, 1966*; *Swets, 1986*; *Wixted, 2020*). Nevertheless, different models can be used in the analysis, that is, models that may or may not be aligned with the actual generative model of the experiment as designed by the researcher or with the internal model that the subject is implicitly using. For example, while classical SDT assumes independence between trials and an experiment may be designed in that way, the subject may assume temporal dependencies between trials. The analysis framework we use here accounts for both these possibilities. A participant may engage in an experiment with unknown subjective costs and false beliefs, as specified in the subjective actor model. Similarly, this analysis framework also allows for the researcher to consider multiple alternative models of behavior and quantify both the uncertainty over individual models' parameters as well as uncertainty over models through Bayesian model comparison.

The limitations of the current method are mainly due to the limitations of the LQG framework. For instance, the assumption that perceptual uncertainty is constant across the whole range of the stimulus domain is adequate for position stimuli, but would not be correct for stimuli that behave according to Weber's law (*Weber, 1834*). This could be addressed using an extension of the LQG to signal-dependent uncertainty (*Todorov, 2005*) to which our inverse optimal control method can be adapted, as we have recently shown (*Schultheis et al., 2021*). Similarly, the assumption of linear dynamics can be overcome using generalizations of the LQG framework (*Todorov and Li, 2005*) or policies parameterized with neural networks (*Kwon et al., 2020*). Finally, assuming independent noise across time steps at the experimental sampling rate of (60 Hz) is certainly a simplifying assumption. Nevertheless, the assumption of independent noise across time steps is very common both in models of perceptual inference as well as in models of motor control, and there is to our knowledge no computationally straightforward way around it in the LQG framework.

Taken together, the current analysis framework opens up the prospect of a wider adoption of continuous psychophysics paradigms in psychology, cognitive science, and neuroscience as it alleviates the necessity of hundreds of repetitive trials with binary forced-choice responses in expert observers, recovering perceptual uncertainties closer to classical paradigms than previous analysis methods. Additionally, it extracts meaningful psychological quantities capturing behavioral variability, effort costs, and subjective perceptual beliefs and provides further evidence for the importance of modeling the behavioral goal and subjective costs in experiments seemingly only investigating perception as perception and action are inseparably intertwined.

## Materials and methods
### LQG control derivations

For the LQG control problem as defined in section 'Computational models of psychophysical tasks,' the *separation principle* between estimation and control holds (*Davis and Vinter, 1985*). This means that the optimal solutions for the estimator $K$ and controller $L$ can be computed independent from one another. Note that this does not mean that actions are independent of perception because $\mathbf{u}_t$ depends on $\hat{\mathbf{x}}_t$.

The optimal estimator is the KF (*Kalman, 1960*)

$$\hat{\mathbf{x}}_t = A\hat{\mathbf{x}}_{t-1} + B\mathbf{u}_{t-1} + K_t\left(\mathbf{y}_t - C\left(A\hat{\mathbf{x}}_{t-1} + B\mathbf{u}_{t-1}\right)\right), \tag{5}$$

where the Kalman gain $K_t = P_t C^T (CP_t C^T + WW^T)^{-1}$ is computed using a discrete-time Riccati equation

$$P_{t+1} = A(P_t - P_t C^T (CP_t C^T + WW^T)^{-1} CP_t)A^T + VV^T \tag{6}$$

initialized with $\hat{\mathbf{x}}_0 = 0$ and $P_0 = VV^T$. Note that our analyses were insensitive to the initialization because state uncertainty and Kalman gains converged after a few time steps, while trials were about 1000 time steps long, which led to a negligible influence of the initial conditions. The optimal controller

$$\mathbf{u}_t = -L_t \hat{\mathbf{x}}_t \tag{7}$$

is the LQR $L_t = (B^T S_{t+1} B + R)^{-1} B^T S_{t+1} A$ and is also computed with a discrete-time Riccati equation

$$S_t = A^T (S_{t+1} - S_{t+1} B(B^T S_{t+1} B + R)^{-1} B^T S_{t+1})A + Q \tag{8}$$

initialized with $S_T = Q$. In principle, all matrices of the dynamical system and thus also $K$ and $L$ could be time-dependent, but since this is not the case in the models considered in this work, we leave the time indices out for notational simplicity. Due to this time invariance of the dynamical system, the gain matrices $K_t$ and $L_t$ typically converge after a few time steps. This allows us to use the converged versions, which we call $K$ and $L$ in our inference method.

## Cross-correlograms

As a summary statistic of the tracking data, we use CCGs as defined by **Mulligan et al., 2013**. The CCG at time lag $\tau$ is the correlation between the velocities of the target $v_t^{(\text{target})}$ at time $t$ and the velocity of the response $v_{t-\tau}^{(\text{response})}$ at time $(t - \tau)$:

$$\rho(\tau) = \frac{\text{cov}(v_t^{(\text{target})}, v_{t-\tau}^{(\text{response})})}{s(v_t^{(\text{target})}) \cdot s(v_{t-\tau}^{(\text{response})})}, \tag{9}$$

where $\text{cov}(x, y)$ and $s(x)$ are the covariance and standard deviation across all time steps and trials. The velocities are estimated from position data via finite differences.

## Likelihood function

Due to the conditional independence assumption implied by the graphical model from the researcher's point of view (**Figure 1F**), the next state $\mathbf{x}_{t+1}$ depends on the whole history of the subject's noisy observations $\mathbf{y}_{1:t}$ and internal estimates $\hat{\mathbf{x}}_{1:t}$. Since, for the researcher, the subject's observations $\mathbf{y}_t$ are latent variables, the Markov property between the $\mathbf{x}_t$ no longer holds and each $\mathbf{x}_t$ depends on all previous $\mathbf{x}_{1:t-1}$, that is, $p(\mathbf{x}_{1:T}|\theta) = \prod_{t=1}^{T-1} p_\theta(\mathbf{x}_t|\mathbf{x}_{1:t-1})$. We can, however, find a more manageable representation of the dynamical system by describing how the $\mathbf{x}_t$ and $\hat{\mathbf{x}}_t$ evolve jointly (**van den Berg et al., 2011**). This allows us to write down a joint dynamical system of $\mathbf{x}_t$ and $\hat{\mathbf{x}}_t$, which only depends on the previous time step:

$$\mathbf{z}_t = \begin{bmatrix} \mathbf{x}_t \\ \hat{\mathbf{x}}_t \end{bmatrix} = F\mathbf{z}_{t-1} + G\omega_t, \ \omega_t \sim \mathcal{N}(0, I). \tag{10}$$

For the derivation of this joint dynamical system and the definitions of $F$ and $G$, see Appendix 1, 'Derivation.' The derivation is extended to account for differences between the true generative model and the agent's subjective internal model in Appendix 1, 'Extension to subjective internal models'.

This allows us to recursively compute the likelihood function in the following way. If we know the distribution $p(\mathbf{z}_t|x_{1:t-1}) = \mathcal{N}(\mu_{t|t-1}, \Sigma_{t|t-1})$, we can compute the joint distribution $p(\mathbf{z}_t, \mathbf{z}_{t+1}|\mathbf{x}_{1:t-1})$ using the dynamical system from (10). Since this distribution is a multivariate Gaussian, we can condition on $\mathbf{x}_t$ and marginalize out the $\hat{\mathbf{x}}_t$ and $\hat{\mathbf{x}}_{t+1}$ to arrive at $p_\theta(\mathbf{x}_{t+1}|\mathbf{x}_{1:t})$. For details, see Appendix 1. The computation of the likelihood function involves looping over a possibly large number of time steps. To make computing gradients w.r.t. the parameters computationally tractable, we use the Python automatic differentiation library `jax` (**Frostig et al., 2018**). Our implementation is available at https://github.com/RothkopfLab/lqg; **Straub, 2022**.

## Bayesian inference

Equipped with the likelihood function derived above, we can use Bayes' theorem to compute the posterior distribution for the parameters of interest:

$$p(\boldsymbol{\theta} \mid \mathbf{x}_{1:T}) \propto p(\mathbf{x}_{1:T}|\boldsymbol{\theta})p(\boldsymbol{\theta}). \tag{11}$$

Since most parameters (except the internal costs $c$) are interpretable in degrees of visual angle, prior distributions $p(\boldsymbol{\theta})$ can be chosen in a reasonable range depending on the experimental setup. We used half-Cauchy priors with the scale parameter $\gamma$ set to 50 for $\sigma$, 25 for $\sigma_p$, 0.5 for $\sigma_m$, and 1 for $c$. We verified that the prior distributions lead to reasonable tracking data using prior predictive checks with the CCGs of the tracking data.

Samples from the posterior were drawn using NUTS (*Hoffman and Gelman, 2014*) as implemented in the probabilistic programming package `numpyro` (*Phan et al., 2019*).

## Model comparison

We compare models using the WAIC (*Watanabe and Opper, 2010*), which approximates the expected log pointwise predictive density. Importantly, unlike AIC it does not make parametric assumptions about the form of the posterior distribution. It can be computed using samples from the posterior distribution (see *Vehtari et al., 2017*, for a detailed explanation):

$$\text{WAIC} = \sum_{i=1}^{N} \log \left( \frac{1}{S} \sum_{s=1}^{S} p\left(\mathbf{x}_i|\boldsymbol{\theta}_s\right) \right) + \sum_{i=1}^{N} V_{s=1}^{S} \left( \log p\left(\mathbf{x}_i|\boldsymbol{\theta}_s\right) \right), \tag{12}$$

where $V_{s=1}^{S}$ is the variance over samples from the posterior. Since WAIC requires individual data points $\mathbf{x}_i$ to be independent given the parameters $\theta$, we define the terms $p\left(\mathbf{x}_i|\boldsymbol{\theta}_s\right)$ as the likelihood of a trial $i$ given posterior sample $s$, which is computed as explained in section 'Likelihood function.' We use the implementation from the Python package `arviz` (*Kumar et al., 2019*).

When comparing multiple models, the WAIC values can be turned into Akaike weights summing to 1

$$w_i = \frac{\exp -0.5\Delta_i}{\sum_j \exp -0.5\Delta_j}, \tag{13}$$

where $\Delta_i$ is the difference between the current model's and the best model's WAIC. The Akaike weights can be interpreted as the relative likelihood of each model.

Note that the assumption of independent noise across time steps might lead to WAIC values that are larger than those obtained under a more realistic noise model involving correlations across time. However, this should not necessarily affect the ranking between models in a systematic way, that is, favoring individual models disproportionately more than others.

## Acknowledgements

We thank Kathryn Bonnen and Lawrence Cormack for sharing their behavioral data and for discussion of continuous psychophysics. Calculations for this research were conducted on the Lichtenberg high-performance computer of the TU Darmstadt. This research was supported by 'The Adaptive Mind,' funded by the Excellence Program of the Hessian Ministry of Higher Education, Science, Research and Art.

## Additional information

### Funding

| Funder | Grant reference number | Author |
|---|---|---|
| Hessian Ministry of Higher Education, Science, Research and Art | The Adaptive Mind | Constantin A Rothkopf |

| Funder | Grant reference number | Author |
|--------|------------------------|--------|

The funders had no role in study design, data collection and interpretation, or the decision to submit the work for publication.

## Author contributions

Dominik Straub, Conceptualization, Software, Formal analysis, Validation, Investigation, Visualization, Methodology, Writing – original draft, Writing – review and editing; Constantin A Rothkopf, Conceptualization, Formal analysis, Supervision, Funding acquisition, Validation, Investigation, Visualization, Methodology, Writing – review and editing

### Author ORCIDs
Dominik Straub ⬤ http://orcid.org/0000-0001-5263-2622
Constantin A Rothkopf ⬤ http://orcid.org/0000-0002-5636-0801

### Decision letter and Author response
Decision letter https://doi.org/10.7554/eLife.76635.sa1
Author response https://doi.org/10.7554/eLife.76635.sa2

## Additional files

### Supplementary files
• MDAR checklist

### Data availability
The current manuscript is a computational study, so no new data have been generated for this manuscript. Modelling code is uploaded at https://github.com/RothkopfLab/lqg, (copy archived at swh:1:rev:58ab4c621081d6eb9eccefd0f3f3c91032ddca38).

The following previously published datasets were used:

| Author(s) | Year | Dataset title | Dataset URL | Database and Identifier |
|-----------|------|---------------|-------------|-------------------------|
| Cormack LK | 2015 | Data from: Continuous psychophysics: Target-tracking to measure visual sensitivity | https://github.com/kbonnen/BonnenEtAl2015_KalmanFilterCode | github.com, BonnenEtAl2015_KalmanFilterCode |
| Knöll J, Pillow JW, Huk A | 2018 | Opticflow tracking data | https://osf.io/h5rxv/ | Open Science Framework, h5rxv |

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

## Appendix 1

### Inverse LQG likelihood

Derivation

We define $\theta$ to be the vector containing any parameters of interest, which influence the LQG system defined above. This could be parameters of the matrices of the dynamical system ($A$, $B$, $V$), the observation model ($C$, $W$), or the cost function ($Q$, $R$). For notational simplicity, we do not explicitly indicate this dependence of the matrices on the parameters. We are interested in the posterior probability

$$p(\theta \,|\, \mathbf{x}_{1:T}) \propto p(\mathbf{x}_{1:T}|\theta)p(\theta). \tag{S1}$$

We start by writing the likelihood using repeated applications of the chain rule:

$$p(\mathbf{x}_{1:T}|\theta) = \prod_{t=1}^{T-1} p_\theta(\mathbf{x}_{t+1}|\mathbf{x}_{1:t}). \tag{S2}$$

The current state $\mathbf{x}_t$ depends on the whole history of previous states $\mathbf{x}_{1:t-1}$ since the internal observations $\mathbf{y}_{1:T}$ and estimates $\hat{\mathbf{x}}_{1:T}$ are unobserved from the researcher's perspective (see *Figure 1F*). The linear Gaussian assumption, however, allows us to marginalize over these variables efficiently. To do this, we start by expressing the current state $\mathbf{x}_t$ and estimate $\hat{\mathbf{x}}_t$ such that they only depend on the previous time step. This idea is based on LQG-MP (*van den Berg et al., 2011*), a method for planning in LQG problems, which computes the distribution of states and estimates without any observations. We start by inserting the LQR control law (*Equation 7*) into the state update (*Equation 1*):

$$\mathbf{x}_{t+1} = A\mathbf{x}_t - BL\hat{\mathbf{x}}_t + V\epsilon_t. \tag{S3}$$

Then, we rewrite the KF update equation (*Equation 5*)

$$\hat{\mathbf{x}}_t \quad = A\hat{\mathbf{x}}_{t-1} + B\mathbf{u}_{t-1} + K(\mathbf{y}_t - C(A\hat{\mathbf{x}}_{t-1} + B\mathbf{u}_{t-1})) \tag{S4}$$

$$= A\hat{\mathbf{x}}_{t-1} - BL\hat{\mathbf{x}}_{t-1} + K(\mathbf{y}_t - C(A\hat{\mathbf{x}}_{t-1} - BL\hat{\mathbf{x}}_{t-1})) \tag{S5}$$

$$= A\hat{\mathbf{x}}_{t-1} - BL\hat{\mathbf{x}}_{t-1} + K(C\mathbf{x}_t + W\eta_t - C(A\hat{\mathbf{x}}_{t-1} - BL\hat{\mathbf{x}}_{t-1})) \tag{S6}$$

$$= A\hat{\mathbf{x}}_{t-1} - BL\hat{\mathbf{x}}_{t-1} + K(C(A\mathbf{x}_{t-1} - BL\hat{\mathbf{x}}_{t-1} + V\epsilon_t) + W\eta_t - C(A\hat{\mathbf{x}}_{t-1} - BL\hat{\mathbf{x}}_{t-1})) \tag{S7}$$

$$= KCA\mathbf{x}_{t-1} + A\hat{\mathbf{x}}_{t-1} - BL\hat{\mathbf{x}}_{t-1} - KCA\hat{\mathbf{x}}_{t-1} + KCV\epsilon_t + KW\eta_t. \tag{S8}$$

Here, we have again inserted the LQR control law (*Equation 7*), then the observation model (*Equation 2*), then the state transition model (*Equation 1*), and finally rearranged the terms. Putting these together, we can write

$$\mathbf{z}_t = \begin{bmatrix} \mathbf{x}_t \\ \hat{\mathbf{x}}_t \end{bmatrix} = \begin{bmatrix} A & -BL \\ KCA & A - BL - KCA \end{bmatrix} \begin{bmatrix} \mathbf{x}_{t-1} \\ \hat{\mathbf{x}}_{t-1} \end{bmatrix} + \begin{bmatrix} V & 0 \\ KCV & KW \end{bmatrix} \begin{bmatrix} \epsilon_t \\ \eta_t \end{bmatrix} \tag{S9}$$

$$= F\mathbf{z}_{t-1} + G\omega_t, \tag{S10}$$

This allows us to recursively compute the likelihood function (*Equation S2*) in the following way. If we know the distribution $p(z_t|x_{1:t-1}) = \mathcal{N}(\mu_{t|t-1}, \Sigma_{t|t-1})$, we can compute the joint distribution $p(\mathbf{z}_t, \mathbf{z}_{t+1}|\mathbf{x}_{1:t-1})$ using the dynamical system from *Equation S10* and the formulas for linear transformations of Gaussian distributions, giving us the mean and covariance matrix

$$\mu_{t,t+1} = \begin{bmatrix} \mu_{t|t-1} \\ F\mu_{t|t-1} \end{bmatrix}, \Sigma_{t,t+1} = \begin{bmatrix} \Sigma_{t|t-1} & \Sigma_{t|t-1}F^T \\ F\Sigma_{t|t-1} & F\Sigma_{t|t-1}F^T + GG^T \end{bmatrix}. \tag{S11}$$

Since this distribution is a multivariate Gaussian, we can marginalize out $\hat{\mathbf{x}}_t$ to obtain $p_{\boldsymbol{\theta}}(\mathbf{x}_t, \mathbf{z}_{t+1}|\mathbf{x}_{1:t-1})$ and then condition on $\mathbf{x}_t$ to arrive at $p_{\boldsymbol{\theta}}(\mathbf{z}_{t+1}|\mathbf{x}_{1:t})$. We do this by partitioning the mean and covariance matrix as

$$\boldsymbol{\mu}_{t,t+1} = \begin{bmatrix} \boldsymbol{\mu}_x \\ \boldsymbol{\mu}_{\hat{x}} \\ F\boldsymbol{\mu}_{t|1:t-1} \end{bmatrix}, \Sigma_{t,t+1} = \begin{bmatrix} \Sigma_{xx} & \Sigma_{x\hat{x}} & \Sigma_{xz} \\ \Sigma_{\hat{x}x} & \Sigma_{\hat{x}\hat{x}} & \Sigma_{\hat{x}z} \\ \Sigma_{zx} & \Sigma_{z\hat{x}} & \Sigma_{zz\cdot} \end{bmatrix} \tag{S12}$$

To marginalize out $\hat{\mathbf{x}}_t$, we simply drop those rows and columns that correspond to $\hat{\mathbf{x}}_t$, yielding

$$p_{\boldsymbol{\theta}}(\mathbf{x}_t, \mathbf{z}_{t+1}|\mathbf{x}_{1:t}) = \mathcal{N}\left(\begin{bmatrix} \boldsymbol{\mu}_x \\ F\boldsymbol{\mu}_{t|1:t-1} \end{bmatrix}, \begin{bmatrix} \Sigma_{xx} & \Sigma_{xz} \\ \Sigma_{zx} & \Sigma_{zz\cdot} \end{bmatrix}\right) \tag{S13}$$

To condition on $\mathbf{x}_t$, we proceed by applying the equations for conditioning in a multivariate Gaussian distribution to obtain

$$p_{\boldsymbol{\theta}}(\mathbf{z}_{t+1}|\mathbf{x}_{1:t}) = \mathcal{N}(\mu_{t+1|1:t}, \Sigma_{t+1|1:t}), \tag{S14}$$

with

$$\mu_{t+1|1:t} = F\boldsymbol{\mu}_{t|1:t-1} + \Sigma_{zx}\Sigma_{xx}^{-1}(\mathbf{x}_t - \boldsymbol{\mu}_x) \tag{S15}$$

and

$$\Sigma_{t+1|1:t} = \Sigma_{zz} - \Sigma_{zx}\Sigma_{xx}^{-1}\Sigma_{xz}. \tag{S16}$$

To obtain the contribution to the likelihood at the current time step $p(\mathbf{x}_{t+1}|\mathbf{x}_{1:t})$, we take $p_{\boldsymbol{\theta}}(\mathbf{z}_{t+1}|\mathbf{x}_{1:t}) = p_{\boldsymbol{\theta}}(\mathbf{x}_{t+1}, \hat{\mathbf{x}}_{t+1}|\mathbf{x}_{1:t})$ and marginalize over $\hat{\mathbf{x}}_{t+1}$. Again, since everything is Gaussian, we can achieve this by simply dropping those rows of the mean and rows and columns of the covariance matrix that correspond to $\hat{\mathbf{x}}_{t+1}$. This concludes our iterative algorithm for computing the likelihood factors. Without loss of generality (since we can subtract the initial time step from the observed trajectories to let them start at zero), we initialize $p(\mathbf{z}_1|\mathbf{x}_0)$ with $\boldsymbol{\mu}_{1|0} = 0$ and $\Sigma_{1|0} = GG^T$.

## Extension to subjective internal models

If we allow the subjective internal model to differ from the true generative model of the experiment, we need to adapt the model equations accordingly. We denote matrices of the true generative model with superscript $g$ and matrices representing the subjective internal model with superscript $s$. The state update model

$$\mathbf{x}_t = A^g\mathbf{x}_{t-1} + B^g\mathbf{u}_{t-1} + V^g\epsilon_t \tag{S17}$$

$$= A^g\mathbf{x}_{t-1} + B^gL\hat{\mathbf{x}}_{t-1} + V^g\epsilon_t. \tag{S18}$$

and the observation model

$$\mathbf{y}_t = C^g\mathbf{x}_t + W^g\boldsymbol{\eta}_t, \tag{S19}$$

remain the same and contain only the matrices of the generative model. The KF update equation, being internal to the subject, is now influenced by several subjective components

$$\hat{\mathbf{x}}_t = A^s\hat{\mathbf{x}}_{t-1} + B^s\mathbf{u}_{t-1} + K(\mathbf{y}_t - C^s(A^s\hat{\mathbf{x}}_{t-1} + B^s\mathbf{u}_{t-1})) \tag{S20}$$

$$= A^s\hat{\mathbf{x}}_{t-1} - B^sL\hat{\mathbf{x}}_{t-1} + K(\mathbf{y}_t - C^s(A^s\hat{\mathbf{x}}_{t-1} - B^sL\hat{\mathbf{x}}_{t-1})) \tag{S21}$$

$$= A^s\hat{\mathbf{x}}_{t-1} - B^sL\hat{\mathbf{x}}_{t-1} + K(C^g\mathbf{x}_t + W^g\eta_t - C^s(A^s\hat{\mathbf{x}}_{t-1} - B^sL\hat{\mathbf{x}}_{t-1})) \tag{S22}$$

$$= A^s \hat{\mathbf{x}}_{t-1} - B^s L \hat{\mathbf{x}}_{t-1} + K(C^g(A^g \mathbf{x}_{t-1} - B^g L \hat{\mathbf{x}}_{t-1} + V^g \epsilon_t) + W^g \eta_t - C^s(A^s \hat{\mathbf{x}}_{t-1} - B^s L \hat{\mathbf{x}}_{t-1})) \tag{S23}$$

$$= KC^g A^g \mathbf{x}_{t-1} + A^s \hat{\mathbf{x}}_{t-1} - B^s L \hat{\mathbf{x}}_{t-1} - KC^s A^s \hat{\mathbf{x}}_{t-1} + K^s C^g V^g \epsilon_t + KW^g \eta_t. \tag{S24}$$

The joint dynamical system of $\mathbf{x}_t$ and $\hat{\mathbf{x}}_t$ is then

$$\mathbf{z}_t = \begin{bmatrix} \mathbf{x}_t \\ \hat{\mathbf{x}}_t \end{bmatrix} = \begin{bmatrix} A^g & -B^g L \\ KC^g A^g & A^s - B^s L - KC^s A^s \end{bmatrix} \begin{bmatrix} \mathbf{x}_{t-1} \\ \hat{\mathbf{x}}_{t-1} \end{bmatrix} + \begin{bmatrix} V^g & 0 \\ KC^g V^g & KW^g \end{bmatrix} \begin{bmatrix} \epsilon_t \\ \eta_t \end{bmatrix} \tag{S25}$$

$$= F\mathbf{z}_{t-1} + G\omega_t, \tag{S26}$$

Then, we proceed by conditioning on the observed states and marginalizing out the subject's internal estimates at each time step as described in the previous subsection.

# Appendix 2

## Models of the tracking task

As explained in section 'Computational models of psychophysical tasks,' we consider a succession of increasingly more complex models, which can be seen as generalizations of the previous ones. In this section, we provide a verbal description of each of the models. *Appendix 2—table 1* contains full definitions of all the matrices needed to define the models.

**Appendix 2—table 1.** Model overview.

| Model | State space | Dynamical system matrices | Cost function | Free parameters |
|---|---|---|---|---|
| Ideal observer | $x_t = \begin{bmatrix} x_t^{(target)} \end{bmatrix}$ | $A = \begin{bmatrix} 1 \end{bmatrix}, B = \begin{bmatrix} 0 \end{bmatrix}, V = \begin{bmatrix} \sigma_{rw} \end{bmatrix}$ <br> $C = \begin{bmatrix} 1 \end{bmatrix}, W = \begin{bmatrix} \sigma \end{bmatrix}$ | - | $\sigma$ |
| Optimal actor | $x_t = \begin{bmatrix} x_t^{(target)} \\ x_t^{(response)} \end{bmatrix}$ | $A = \begin{bmatrix} 1 & 0 \\ 0 & 1 \end{bmatrix}, B = \begin{bmatrix} 0 \\ dt \end{bmatrix}, V = \begin{bmatrix} \sigma_{rw} & 0 \\ 0 & \sigma_m \end{bmatrix}$ <br> $C = \begin{bmatrix} 1 & 0 \\ 0 & 1 \end{bmatrix}, W = \begin{bmatrix} \sigma & 0 \\ 0 & \sigma_p \end{bmatrix}$ | $Q = \begin{bmatrix} 1 & -1 \\ -1 & 1 \end{bmatrix}, R = \begin{bmatrix} 0 \end{bmatrix}$ | $\sigma, \sigma_m, \sigma_p$ |
| Bounded actor | $\mathbf{x}_t$ as above <br> $\mathbf{x}_t$ as above | $A, B, V, C, W$ as above <br> $A, B, V, C, W$ as above | $Q$ as above, <br> $Q, R$ as above | $\sigma, \sigma_m, \sigma_p, c$ |
| Subjective actor | $x_t^{(s)} = \begin{bmatrix} x_t^{(target)} \\ x_t^{(response)} \\ v_t \end{bmatrix}$ | $A^s = \begin{bmatrix} 1 & 0 & dt \\ 0 & 1 & 0 \\ 0 & 0 & 1 \end{bmatrix}, B^s = \begin{bmatrix} 0 \\ dt \\ 0 \end{bmatrix}, V^s = \begin{bmatrix} \sigma_s & 0 & 0 \\ 0 & \sigma_m & 0 \\ 0 & 0 & \sigma_v \end{bmatrix}$ <br> $C^s = \begin{bmatrix} 1 & 0 & 0 \\ 0 & 1 & 0 \end{bmatrix}, W^s = W$ | $Q^s = \begin{bmatrix} 1 & -1 & 0 \\ -1 & 1 & 0 \\ 0 & 0 & 0 \end{bmatrix}$ <br> $R^s = R$ | $\dot{\sigma}, \sigma_m, \sigma_p, c, \sigma_s, \sigma_v$ |

## Ideal observer (KF)

For the KF model, the state is simply the position of the target, $x_t = \begin{bmatrix} x_t^{(target)} \end{bmatrix}$. The subject's cursor position is taken as the internal estimate of the state $\hat{x}_t$. The likelihood of the observed data $(\mathbf{x}_{1:T}, \hat{\mathbf{x}}_{1:T})$ can be computed as above (Appendix 1), but since the model does not explicitly include actions and instead treats the internal estimate as the subject's cursor position, we need to condition on $\mathbf{x}_{1:T}$ and $\hat{\mathbf{x}}_{1:T}$ at every time step.

## Optimal and bounded actor

The models based on LQG control explicitly include the subject's cursor position as a part of the state. In the simplest case, the state is two-dimensional and contains the position of the target stimulus and the subject's position of the response: $x_t = \begin{bmatrix} x_t^{(target)} & x_t^{(response)} \end{bmatrix}^T$. The continuous psychophysics experiments considered here used target stimuli moving according to a simple Gaussian random walk.

Accordingly, the model may assume that the target only moves because of the random walk with standard deviation $\sigma_{rw}$ and that the response is influenced by the agent's control input and by action variability $\sigma_m$. The agent observes the target and response position separately, with independent observation noise in both dimensions ($\sigma$ for the target and $\sigma_p$ for the response). The behavioral cost of actions is penalized by the parameter $c$. For the optimal actor without internal costs, we set $c = 0$, while for the bounded actor $c$ is a free parameter.

## Subjective actor

The target's position in the tracking task by *Bonnen et al., 2015* is on a random walk,

$$x_{t+1}^{(target)} = x_t^{(target)} + \sigma_{rw}\epsilon_t, \ \epsilon_t \sim \mathcal{N}(0,1). \text{(S27)}$$

The subjective actor may assume that the target dynamics differ from this true random walk in that there is still a random walk on the target's position, but the standard deviation can be different from the true standard deviation. In addition, the subjective actor may assume that in addition to

the target position's random walk, the target's velocity can change over time according to its own random walk,

$$x_{t+1}^{(\text{target})} = x_t^{(\text{target})} + dt \cdot v_t + \sigma_s \epsilon_t, \ \epsilon_t \sim \mathcal{N}(0,1) \tag{S28}$$

$$v_{t+1} = v_t + \sigma_v \xi_t, \ \xi_t \sim \mathcal{N}(0,1). \tag{S29}$$

Thus, for the subjective actor, the actual matrices of the generative model remain identical to the basic model, but the agent assumes a different dynamical system, in which the target has an additional dimension representing its velocity $x_t = \begin{bmatrix} x_t^{(\text{target})} & x_t^{(\text{response})} & v_t \end{bmatrix}^T$. The agent still receives an observation of the positions only, with independent observation noise on both dimensions. The parameters $\sigma_s$ and $\sigma_v$ represent the subject's belief about the standard deviation of the target's position and velocity. The matrices of the subjective part of the model are changed to represent the subjective dynamics described here verbally (see *Appendix 2—table 1*).

# Appendix 3

## Simulations

### Parameter recovery

To evaluate the inference method, 200 sets of parameters for the subjective actor model were sampled from the following uniform distributions:

$$
\begin{aligned}
c &\sim U(0.05, 2) \\
\sigma &\sim U(0.5, 100) \\
\sigma_m &\sim U(0.05, 1) \\
\sigma_p &\sim U(0.5, 50) \\
\sigma_s &\sim U(0.1, 2) \\
\sigma_v &\sim U(0.1, 2).
\end{aligned}
$$

For each set of parameters, we simulated datasets with the following number of trials: 10, 25, and 50 trials. Each trial was 720 time steps long. For each dataset, four Markov chains with 7500 samples each were drawn from the posterior distribution, after 2500 warm-up steps.

### Model recovery

To investigate how different models behave when fit to tracking data generated from one of the other models, we performed a model recovery analysis. To this end, 20 different values for the parameters $c$ and $\sigma_m$ were sampled from uniform distributions defined above. The values of the other model parameter were fixed at $\sigma = 6$, $\sigma_p = 1$, $\sigma_s = 0.1$ and $\sigma_v = 10$. For each set of parameters, we simulated datasets of 50 trials with 500 time steps per trial from the bounded actor model and the subjective model. Then, we estimated fits for each of the four different models presented in *Figure 2* to the tracking data. We ran four chains with 5000 samples each after 2000 warm-up steps.

## Appendix 4

### Experiments

We reanalyze data from three previous publications (*Bonnen et al., 2015*; *Bonnen et al., 2017*; *Knöll et al., 2018*). In this section, we describe our data analysis procedures for these datasets. For detailed information about the experiments, we refer to the original publications.

### Continuous psychophysics

First, we reanalyze data from three subjects in two previously published experiments (*Bonnen et al., 2015*): a position discrimination and a position tracking task. The stimuli were 'Gaussian blobs,' two-dimensional Gaussian functions on Gaussian pixel noise, which changed from frame to frame. The visibility of the stimuli was manipulated via the standard deviation of the Gaussian blobs, with larger standard deviations leading to lower luminance increments and thus to a weaker signal. The standard deviations were 11, 13, 17, 21, 25, and 29 arcmin.

In the discrimination experiment, position discrimination thresholds for each blob width were measured in a two-interval forced-choice (2IFC) task. We fit cumulative Gaussian psychometric functions with lapse rates in a Beta-Binomial model to the discrimination performance using `psignifit` (*Schütt et al., 2015*). We use the posterior mean of the standard deviation of the Gaussian psychometric function as our estimate of the perceptual uncertainty in the 2IFC task.

In the tracking experiment, the same visual stimuli moved on a random walk with a standard deviation of one pixel (1.32 arcmin of visual angle) per frame at a frame rate of 60 Hz. Subjects were instructed to track the target with a small red mouse cursor. Each subject completed 20 trials for each blob width, with each trial lasting 1200 time steps (20 s). As in *Bonnen et al., 2015*, the response time series were shifted by $\tau = 12$ time steps w.r.t. the target time series to account for a constant time lag. Instead, one could explicitly account for the time lag in the observation function of the model by extending the state space to include the $\tau$ previous time steps, but we found no appreciable differences in the model fits. We obtain posterior mean perceptual uncertainty estimates using the methods described in sections 'Likelihood function' and 'Bayesian inference'. Since there are two stimulus presentations in a 2IFC task and we are interested in perceptual uncertainty for a single stimulus presentation, we divided the estimates of the perceptual uncertainty by $\sqrt{2}$ to obtain single-interval sensory uncertainties. This corresponds to the assumption that the subject judges the difference between the stimuli observed in the two intervals. Although this assumption is debated and empirical deviations in both directions were observed (*Yeshurun et al., 2008*), as discussed in the main text, this assumption is a natural starting point when converting between 2IFC and single-interval perceptual uncertainties. Because the stimuli in the 2IFC task were shown for 15 frames and the computational models of the tracking task operate on single frames, we apply a second correction factor to the perceptual uncertainty estimates. Assuming optimal integration of the Gaussian uncertainty across frames, the 2IFC sensory uncertainty estimates were multiplied by a constant factor of $\sqrt{15}$.

### Motion tracking

We reanalyzed the data from experiment 2 of *Bonnen et al., 2017*, in which subjects tracked a circular target on a gray background with their cursor. Instead of the computer mouse, the cursor was controlled used a Leap Motion hand tracking device (Leap Motion, San Francisco, USA). There were five different standard deviations for the target's random walk. Each subject completed 20 trials per standard deviation in randomly interleaved blocks of 10 trials. As above, the response time series were shifted by 12 time steps w.r.t. the target time series.

### Gaze tracking

We reanalyzed the data from an experiment on gaze tracking (*Knöll et al., 2018*), in which two macaques (M1 and M2), one human participant (H), and a marmoset (C) tracked the center of an optical flow field displayed on a screen, while their eye movements were recorded. In this experiment, the temporal shift applied to the response time series was 10 time steps, which was determined from the CCGs.

## Appendix 5

### Supplementary results

#### Cross-validation

For the experiment from *Bonnen et al., 2017*, in addition to the model fits on the data from all conditions, we performed a cross-validation evaluation for the tracking experiment with different random walk conditions. In experiment 2 from the study by *Bonnen et al., 2017*, there were five different standard deviation conditions for the target's random walk, which were presented in blocks (see Appendix 4, 'Motion tracking'). Specifically, we fit all models to the data in a leave-one-condition-out cross-validation scheme, that is, we fit the model on data from four conditions and evaluate it on the remaining condition, and repeated this procedure for each condition. As a metric for evaluation, we compute the log predictive posterior density relative to the best performing model ($\Delta \log p$). It was higher for the subjective model than for the other models in all but one condition of one participant (*Appendix 5—figure 1A*). To check which model yields the most reliable estimates of perceptual uncertainty, we look at the posterior mean estimates of $\sigma$ (*Appendix 5—figure 1B*). The subjective model's estimates are most consistent across cross-validation conditions, with an average standard deviation of 0.25 arcmin, compared to 0.60 for the basic LQG and 0.52 for the KF. This suggests that the model recovers the subjects' perceptual uncertainty more reliably than the alternative models.

#### Application to eye-tracking data

We reanalyze data from the experiment described in Appendix 4, 'Gaze tracking.' Since the generative model of the target's random walk was slightly different compared to the other experiments, the model was adapted accordingly.

The target dynamics has a velocity component that depends on the current position, such that the target tends toward the center of the screen. The matrices of the dynamical system and cost function are

$$A = \begin{bmatrix} 1 & 0 & 1 \\ 0 & 1 & 0 \\ -k & 0 & \lambda \end{bmatrix}, \ B = \begin{bmatrix} 0 \\ dt \\ 0 \end{bmatrix}, \ V = \mathrm{diag}(0, \sigma_m, \sigma_w), \ C = \begin{bmatrix} 1 & 0 & 0 \\ 0 & 1 & 0 \end{bmatrix}, \ W = \mathrm{diag}(\sigma, 0), \tag{S30}$$

$$Q = \begin{bmatrix} 1 & -1 & 0 \\ -1 & 1 & 0 \\ 0 & 0 & 0 \end{bmatrix}, \ R = \begin{bmatrix} c \end{bmatrix} \tag{S31}$$

Thus, the agent's estimate is also three-dimensional, that is, the agent estimates a velocity from their observations of the position.

As for the other experiments, we fit all four models to the data and computed WAIC as a model comparison metric (*Appendix 5—figure 2*). Again, the behavioral data is better accounted for by the LQG models compared to the KF model. As in the other analyses of experimental data, the bounded actor models with internal costs better account for the data than the optimal actor model.

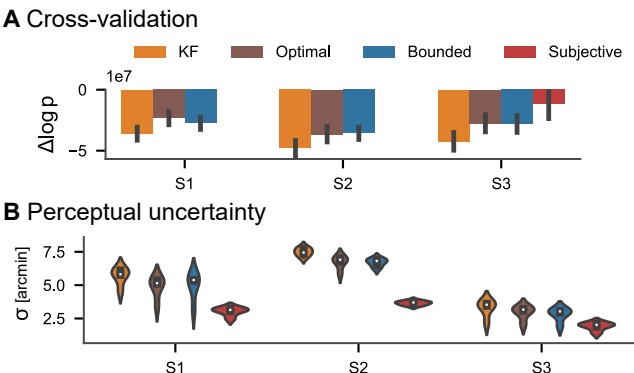

**Appendix 5—figure 1.** Cross-validation of estimated parameters. (**A**) Difference in log-posterior relative to the best model per left-out condition from the experiment by *Bonnen et al., 2017*. Each model was fitted on the remaining four conditions as described in Appendix 5. In all but one condition of one participant, the model with a subjective component accounts best for the data. Error bars indicate 95% CIs across the 5 cross-validation runs. (**B**) Perceptual uncertainty (posterior mean) estimates across cross-validation conditions.

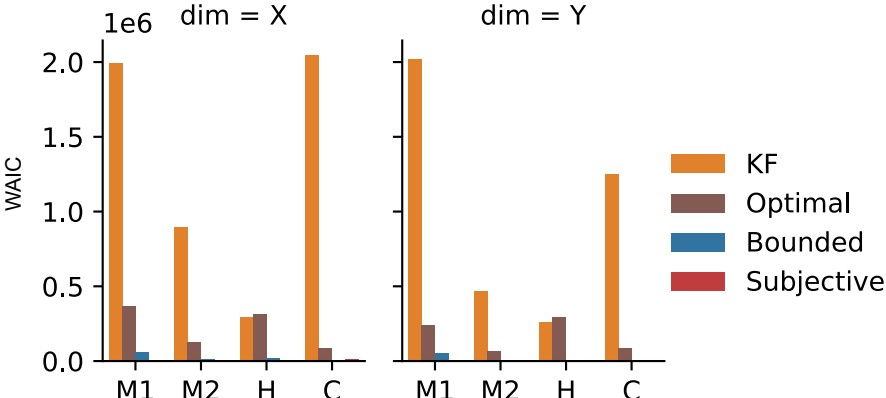

**Appendix 5—figure 2.** Model comparison on eye-tracking data. Widely applicable information criterion (WAIC) on our four models for the eye-tracking data, fit separately to the X and Y dimension.

