## [Editor Report]

This important article presents a Bayesian model framework for estimating individual perceptual uncertainty from continuous tracking data – a powerful and exciting alternative to traditional binary-choice psychophysical experiments. The model takes into account motor variability, action cost, and possible misestimation of the generative dynamics. The analyses provide compelling evidence for the framework. The article is clearly written and provides a didactic resource for students wishing to implement similar models on continuous action data.

---

## [Decision Letter]

**Decision letter after peer review:**

Thank you for submitting your article "Putting perception into action: Inverse optimal control for continuous psychophysics" for consideration by *eLife*. Your article has been reviewed by 3 peer reviewers, including Jörn Diedrichsen as Reviewing Editor and Reviewer #1, and the evaluation has been overseen by Joshua Gold as the Senior Editor.

The reviewers have discussed their reviews with one another, and the Reviewing Editor has drafted this to help you prepare a revised submission. Overall, all reviewers thought that the paper has the potential to become a very valuable contribution to the field. The revisions focus mainly on the clarity of the submission. The extensive original reviews are attached below and should be responded to in detail.

Essential revisions:

1) The assumption of independence of motor noise across observations (reviewer #1, comment #1) is likely problematic in terms of obtaining valid WAIC values – and this problem should be addressed.

2) The technical exposition in the introduction is sometimes confusing and the clarity of the explanations should be improved. The specific comments provide a detailed list of constructive suggestions here.

3) The discussion of the previous literature is not always as balanced and would be ideal (see reviewer #2)

4) For a tools and resource paper, I would hold that the associated code and documentation should be visible to the reviewers at the point of submission, so the quality of the online resource can be reviewed together with the paper. So I would ask you to provide a working link with the revision of the paper.

*Reviewer #1: All comments in public review.*

*Reviewer #2 (Recommendations for the authors):*

Specific Remarks:

– There are passages that read like poorly grounded indictments of well-established literature. Supplementary Section A, which functions as support for claims in the main text and the figure 3 caption, references a study by Yeshurun et al., (2008). The authors of the present paper write that Yeshurun et al., 'investigated the validity of the theoretical result (Green and Swets, 1966), that the forced-choice sensitivity d'FC differs from the sensitivity obtained through an equivalent Yes-No task d'YN by a factor of √2, i.e. d'FC = √2d'YN. Apart from running a serious of experiments targeted at evaluating the validity of this theoretical result, the authors reported the actual values of this factor based on extensive literature review involving numerous studies investigating perceptual sensitivities across sensory modalities. Indeed, the literature review collected a broad range of deviations from the theoretical value.'

The authors then go on to specifically reference two of the papers cited by Yeshurun (2008). One of these papers (Jesteadt and Bilger, 1974) reported that d-prime estimated with a forced-choice task was better by a factor of 2.1 than performance in an equivalent yes-no task. The other (Markowitz and Swets, 1967) reported d-prime estimated with a forced-choice task was better by only 1.15. These two studies deviated from the predicted 1.4 improvement by factors of 1.5x and 0.8x, respectively.

The logic of the passage is problematic, and the passage seems only tenuously connected to the aims of the paper. In my view, I think the supplementary passage and the connected threads in the main text are harmful to the paper's messaging and should be removed. Here is why: The theoretical result relating sensitivity in forced-choice and yes-no tasks as reported by Green and Swets (1966) is easy to derive in a few lines and is most certainly valid, presuming that certain assumptions hold. The authors can of course fairly question whether these assumptions hold in certain experiments (as did Yeshurun et al., 2008), but experiments do not, and could not, evaluate the 'validity of this theoretical result'. The theoretical result stands on its own. The authors' writing seems to conflate a lack of empirical variability (e.g. due to measurement error, assumptions not holding) with a lack of theoretical validity. I thought, initially, that this could perhaps be explained away by inattentive writing.

But the authors' then use the range of deviations aforementioned deviations-the 1.5x deviation reported by Jesteadt and Bliger was on auditory frequency and intensity discrimination; the 0.8x deviation reported by Markowitz and Swets, was on sound detection-to suggest that the observed 1.36x deviations between tracking- and forced-choice-based estimates (from Model 4 fits) are within an established range of deviations in the literature. The relevance of these numbers to the present work is dubious at best. Consider, for example, if some other pair of studies on gustatory sodium discrimination or detection had reported deviations ranging from 0.25x to 8x the theoretically predicted values. Would that somehow be relevant to the interpretation of the current work? I am not sure why the authors thought this a valuable addition to the paper. They seem to be using these studies to suggest that because other studies have deviated from predictions (whatever the causes) that discrepancies in the present study should be thought of as small. It all feels forced. After writing out these notes, I was concerned that perhaps I had been making too much of it. But the next issue has a similar flavor.

– The manuscript appears both to misleadingly describe findings in Bonnen et al., (2015), and to self-contradict its own assertions regarding those findings. The authors several times write that Bonnen et al.,'s tracking-based estimates of position uncertainty (i.e. position discrimination thresholds) are an order of magnitude (i.e. ~10x) higher than corresponding forced-choice-based estimates. The authors contrast this discrepancy with their own results, saying that they 'infer values that differ by a factor of 1.36 on average'. It is true that the raw estimates of Bonnen et al., reported tracking-based estimates that were ~10x larger. But that was before Bonnen et al., took several obvious factors into account (e.g. the benefit of integrating across 15 frames of each stimulus presentation, the predicted improvement in sensitivity in 'two-look' forced choice tasks). After taking these factors into account (as it is appropriate to do), the ~10x difference between the tracking- and forced-choice-based was reduced to a ~2x difference. In the supplement, the authors acknowledge that Bonnen et al., took these factors into account, but their writing in the abstract, introduction, and discussion (references to order of magnitude differences) reads as if they are unaware. I am unclear about the reasons for this discrepancy, but it should be corrected.

Further, on page 5, left column the authors report that estimates of perceptual uncertainty of target position from the Kalman Filter model differ from the forced-choice-based estimates by approximately 2x. They write: 'The average factor between the posterior mean perceptual uncertainty in the continuous task and the 2IFC task in the 5 higher contrast conditions is 1.20 for the LQG model, while it is 1.73 for the KF. Only in the lowest contrast condition, it increases to 2.20 and 2.93 arcmin, respectively.'. This difference is in line with the conclusions of Bonnen et al., and does not square with the assertion that tracking-based estimates were off by an order of magnitude (~10x). Given that the current Kalman filter analysis essentially replicates the analysis performed by Bonnen et al., these two assertions (10x differences vs 2x differences) cannot both be correct. Again, in keeping with the first point above, all this seems like a forced attempt to convince the reader that observed deviations are small between the tracking- vs. forced-choice-based estimates of perceptual uncertainty about target position. But it seems unnecessary. Model 4 does a more accurate job of estimating perceptual uncertainty about target position than simpler models (and seems to provide a closer match to forced-choice-based estimates). Let those results stand on their own.

– For a computational paper, in which the primary contribution is to develop methods for data analysis and parameter estimation, there should be substantially more discussion of which parameters have their values trade off of one another. For example, the authors have taken the trouble visualize a posterior distribution over the parameters (Figure 3A). They should help the reader develop an intuition for why the posterior has the structure that it does. All models other than the Kalman filter model include perceptual uncertainty about target position (\σ), perceptual uncertainty about cursor position (\σ_p), and motor variability. All of these quantities affect the reliability with which discrepancies between target and cursor position can be evaluated, and should have similar effects on performance. The figure shows that these parameters trade-off with each other (strong negative correlations in the posterior distributions over the model parameters). The authors should include some discussion of these trade-offs, helping to provide the reader intuition for why these parameters trade-off in the way that they do. Similarly for the positive correlation between motor variability (\σ_m) and perceptual uncertainty about cursor position.

– Regarding their analysis of simulated data, the authors report that attempts to infer the value of the target position uncertainty (\σ) from Model 4 were accurate (Figure 3C). It would be helpful for the authors to describe in more detail the parameters of the simulation. Specifically, in would be useful to explain and provide intuition for how, if the observer had a mistaken belief regarding the drift dynamics, it is possible to accurately infer the value of perceptual uncertainty about target position. A naïve reader might presume that, because there is an infinite number of pairs of position uncertainty (\σ) and presumed drift variance (\σ_s) that determine the Kalman gain, that \σ could not in fact be accurately estimated if \σ_s does not equal the true drift parameter (\σ_rw). Please explain for the reader how this works.

Tone

The current paper describes shortcomings of the Bonnen et al., (2015) data analyses. Many of these shortcomings were explicitly acknowledged by Bonnen et al., It would help the tone and tenor of the manuscript if, when the current authors describe the acknowledged shortcomings, the current authors cite Bonnen et al., (2015).

Examples include:

– "A model, which only considers the perceptual side of the task, will therefore tend to overestimate perceptual uncertainty because these additional factors get lumped into perceptual model parameters, as we will show in simulations."

– "A model without these factors needs to attribute all the experimentally measured behavioral biases and variability to perceptual factors, even when they are potentially caused by additional cognitive and motor processes."

The current paper makes a nice contribution in demonstrating these points quantitatively. But, in places (and in keeping with the flavor of remarks above), the writing seems concerned that readers might not recognize the paper's unique contribution. I think the paper's contribution is clear. And I think that their paper will read better, and leave a better impression, if it looks for ways to portray previous work in a more generous light.

Specific Comments

– Experimentor/Participant distinction

The experimenter has access to the true target position and the true cursor position is available. The experimental participant him/herself has access only to the estimates of target and cursor positions. The article should more explicitly discuss this issue in the main text. Figure 2 and the Supplemental derivations allude to it, but it should be discussed more prominently.

– Explanation of matrix values.

The authors should explain why the B=[0 dt]' parameter takes on the particular values that it does. Is the implication that the control action is best understood as a 'rate of change' so that result matrix-multiplying the control action with B is a position? Please explain. The value of the non-zero element of B trades off perfectly with the square root (?or similar?) of the movement cost parameter 'c'. So it would be valuable for the authors to explain why they chose to set B to have the value it does.

Telegraphic mathematical development.

In the supplement, the mathematics associated with transitioning between Supplemental Equations S16-S18 needs to be expanded. Conditioning the nD Gaussian on x_t and marginalizing out x_hat_t are two separate steps, but the manner in which the passage is written encourages the reader to presume that they are a single step. Too much analytical work is foisted on the reader if he/she wants to carefully follow the derivation along. The derivation is correct (I worked it out myself), but it should be laid out more explicitly. Further, the final expression for the likelihood of the state (x_{t+1}) on time step t+1 should be provided. This likelihood is straightforwardly obtained by marginalizing out the estimate (x^{hat}_{t+1}) of the (2-vector) state, but it would be nice for the reader if the final expression was actually made explicit for the reader.

Subjective model of state dynamics

The authors should make explicit in equation form the fallacious state dynamics that are assumed by the subjective observer. The authors describe it in words, but equations will prevent any uncertainty that the description may produce. Something like:

position walk: p_{t+1} = p_t + eps; where eps ~ N(0, σ_{rw})

velocity walk: v_{t+1} = v_t + eta; where eta ~ N(0, σ_v )

x_{t+1} = p_{t+1} + sum( v_{ 0:(t+1) } ) where the sum goes from 0 to t+1

Subjective model of estimate of velocity

Also, it is not made completely clear how the subjective observer's fallacious estimate of velocity is computed, how it resides in the model, whether the state is expressed as a 2-vector [ x_t x_p ] or a 3-vector [ x_t x_p v_t ], and whether the estimate of the state is expressed as a two-vector [ xhat_t xhat_p ] or a 3-vector [ xhat_t xhat_p vhat_t ]. An expanded discussion of these issues should be included. The expressions are correct but as they stand, but the mathematical development and discussion of these points are a bit too telegraphic.

Typo in the supplementary equation.

I believe that there is a typo in equation S2 of the supplement where the Ricatti equation is laid out. The term with the inverse currently reads ( C*P*C' )^-1 whereas this term in standard expressions for the Ricatti equation should be ( C*P*C' + W*W' )^-1. This term follows the same form as the inverse term in the expression for the Kalman gain (see the line above) which reflects the total covariance ( prior + observation covariances ).

*Reviewer #3 (Recommendations for the authors):*

The paper is already in a very polished state. I have only a few comments about clarity and completeness.

– I found myself getting confused while reading the paper about what parameters were actually involved in each model (e.g. pg 3: "the KF model has only one parameter: perceptual uncertainty σ"). My recommendation to the authors is to move equations 1-4 out of the Methods section and into the main text. Personally I would consider this part of the results (ie. "what are the models we're using"?) and so I would rather see this presented pedagogically within the main paper. Keep implementation details or other technical issues in the Methods, but present the models themselves in the Results section. Just a recommendation, but that's my 2 cents!

– This leads to a related note on clarity: it's not entirely clear to me which parameters are being fit in each model. Presumably you're not fitting the C matrix in equation 2? What about B in equation 1? Come to think of it, I'm not sure where the cursor fits in, allowing for uncertainty in cursor position). (Does that become part of the state vector x_t? Please unpack this more clearly so that we can understand what these equations correspond to for each of the models discussed in the paper!

– "Our implementation will be made available on github" -- I want to note that this should be a requirement for acceptance! (i.e., the code should be posted before the paper is accepted).

– The name "bounded actor" seems like a poor one, since there aren't any bounds on the actor. (There are just costs). Figure 2 C refers to it as "bounded actor with behavioral costs" -- personally I would keep the "with behavioral costs" and drop the "bounded".

– One other point that would be worth making: you could also formulate a "with subjective belief" version of any of the simpler models. Even in the simple KF model, you could allow for mismatch between the true dynamics and the subject's belief about the dynamics. I think it's probably ok to leave the model comparisons as is, since obviously it becomes a bit messy if we have to include "optimal" and "subjective" versions of each of the models, but you should at least mention somewhere in the paper that this can be used to improve the accuracy (in terms of matching observer behavior) for any of the models.

– Can you say anything about the discrete time assumptions of the model? i.e., how would you expect the model parameters (or the accuracy of the fit) to change if you switched to 120 Hz frame rate or 20 Hz frame rate?

– Finally, a small note on grammar/usage: the paper frequently uses "… , which…" in cases where "that" (with no comma). e.g., in the abstract: "recover perceptual thresholds, which are one order of magnitude larger compared to equivalent traditional psychophysical experiments". This makes it sound like you're providing a definition of threshold, rather than describing a property of the recovered thresholds. More correct would be: "We recover perceptual thresholds that are one order of magnitude larger than …". I found this issue repeatedly in the text. You could basically look for nearly every occurrence of ", which" and replace by "that".

---

## [Author Response]

Essential revisions:1) The assumption of independence of motor noise across observations (reviewer #1, comment #1) is likely problematic in terms of obtaining valid WAIC values – and this problem should be addressed.2) The technical exposition in the introduction is sometimes confusing and the clarity of the explanations should be improved. The specific comments provide a detailed list of constructive suggestions here.3) The discussion of the previous literature is not always as balanced and would be ideal (see reviewer #2)4) For a tools and resource paper, I would hold that the associated code and documentation should be visible to the reviewers at the point of submission, so the quality of the online resource can be reviewed together with the paper. So I would ask you to provide a working link with the revision of the paper.

Yes, thank you for summarizing the main points by the reviewers. We address all of these points in the responses to individual reviewers in the following. The link to the code associated with our manuscript is https://github.com/RothkopfLab/lqg and has been added in the manuscript.

Reviewer #1: All comments in public review.

Responses provided in response to public review.

Reviewer #2 (Recommendations for the authors):Specific Remarks:– There are passages that read like poorly grounded indictments of well-established literature. Supplementary Section A, which functions as support for claims in the main text and the figure 3 caption, references a study by Yeshurun et al., (2008). The authors of the present paper write that Yeshurun et al., 'investigated the validity of the theoretical result (Green and Swets, 1966), that the forced-choice sensitivity d'FC differs from the sensitivity obtained through an equivalent Yes-No task d'YN by a factor of √2, i.e. d'FC = √2d'YN. Apart from running a serious of experiments targeted at evaluating the validity of this theoretical result, the authors reported the actual values of this factor based on extensive literature review involving numerous studies investigating perceptual sensitivities across sensory modalities. Indeed, the literature review collected a broad range of deviations from the theoretical value.'The authors then go on to specifically reference two of the papers cited by Yeshurun (2008). One of these papers (Jesteadt and Bilger, 1974) reported that d-prime estimated with a forced-choice task was better by a factor of 2.1 than performance in an equivalent yes-no task. The other (Markowitz and Swets, 1967) reported d-prime estimated with a forced-choice task was better by only 1.15. These two studies deviated from the predicted 1.4 improvement by factors of 1.5x and 0.8x, respectively.The logic of the passage is problematic, and the passage seems only tenuously connected to the aims of the paper. In my view, I think the supplementary passage and the connected threads in the main text are harmful to the paper's messaging and should be removed. Here is why: The theoretical result relating sensitivity in forced-choice and yes-no tasks as reported by Green and Swets (1966) is easy to derive in a few lines and is most certainly valid, presuming that certain assumptions hold. The authors can of course fairly question whether these assumptions hold in certain experiments (as did Yeshurun et al., 2008), but experiments do not, and could not, evaluate the 'validity of this theoretical result'. The theoretical result stands on its own. The authors' writing seems to conflate a lack of empirical variability (e.g. due to measurement error, assumptions not holding) with a lack of theoretical validity. I thought, initially, that this could perhaps be explained away by inattentive writing.But the authors' then use the range of deviations aforementioned deviations-the 1.5x deviation reported by Jesteadt and Bliger was on auditory frequency and intensity discrimination; the 0.8x deviation reported by Markowitz and Swets, was on sound detection-to suggest that the observed 1.36x deviations between tracking- and forced-choice-based estimates (from Model 4 fits) are within an established range of deviations in the literature. The relevance of these numbers to the present work is dubious at best. Consider, for example, if some other pair of studies on gustatory sodium discrimination or detection had reported deviations ranging from 0.25x to 8x the theoretically predicted values. Would that somehow be relevant to the interpretation of the current work? I am not sure why the authors thought this a valuable addition to the paper. They seem to be using these studies to suggest that because other studies have deviated from predictions (whatever the causes) that discrepancies in the present study should be thought of as small. It all feels forced. After writing out these notes, I was concerned that perhaps I had been making too much of it. But the next issue has a similar flavor.

We were not as careful in our wording as we should have been. Of course, experiments finding empirical differences in perceptual uncertainties between psychophysical tasks deviating from the differences predicted by SDT cannot invalidate the theoretical result of a sqrt(2) factor between 2-AFC and Y-N experiments based on signal detection theory. We fully agree on that. We have therefore replaced phrases like “investigated the validity of the theoretical result” with more adequate wording such as “empirically tested the theoretical result”.

The point we were trying to make by citing Yeshurun et al., (2008) is that, irrespective of the sensory modality, it is very common to find discrepancies between the predictions of SDT across different psychophysical tasks and the empirical data across different psychophysical tasks. How large are these discrepancies? We turned to Yeshurun et al., (2008) to provide a quantitative value for these discrepancies.

This is not an “indictment” of “well-established literature” but a scientific fact. The first sentence of our abstract should leave no doubt that no “indictments” are involved here. If there are other studies showing that empirical data in different psychophysical tasks agree with the theoretical predictions, we will be happy to cite them.

Why are we trying to make this point?

Because it is not at all straightforward how to interpret these discrepancies, scientifically. One common criticism towards continuous psychophysics tracking experiments that we have encountered repeatedly is that they introduce additional uncontrolled factors and therefore yield biased estimates of sensory uncertainty. Classic psychophysics paradigms such as 2-AFC and Y-N experiments together with SDT, on the other hand, are said to provide unbiased and consistent estimates of sensory uncertainty. But, Yeshurun et al., (2008) have carefully reviewed the literature and carried out experiments to establish that, empirically, there are significant discrepancies across numerous classical tasks. Jäkel and Wichmann (2006) also report systematic discrepancies between the empirical results and the results expected by SDT across experiments. Numerous studies also have reported pervasive systematic sequential effects across classic psychophysical experiments, a fact that has attracted a lot of interest recently. Thus, in our view, there is enough empirical evidence to support the view that the analysis of behavioral data obtained through classical psychophysical experiments with SDT is useful, but that there are almost always factors in the experiments that are not perfectly captured by models based on SDT, such as attention, memory, intrinsic costs, beliefs on the side of the subject differing from the true task statistics employed by the researcher, learning, and potentially more.

This applies to trial-based tasks as well as continuous tasks. In our models, we have included some of these additional factors for the tracking task, and shown that this helps decrease the discrepancy between the sensory uncertainty inferred from the tracking task and the 2-AFC task. Therefore, it is important to us to highlight that the remaining discrepancy is not a particularity of continuous psychophysics as a new experimental paradigm, but typical when comparing different psychophysical tasks.

We agree that the concrete values in different sensory modalities derived from the studies by Jestead and Bilger (1974) and Markowitz and Swets (1967) may not be as relevant to the visual position discrimination and tracking tasks conducted by Bonnen et al., (2015). We have therefore removed references to these concrete values from the Results section (Figure 4A, where we now plot posterior CIs instead), the Discussion section, and the supplementary material. In the discussion, we instead cite Yeshurun et al., (2008) simply as evidence for the discrepancies between empirical measurements and the predictions by SDT across tasks.

As requested by the reviewer, we have removed the passage in the supplementary material and revised the discussion of this point in the Discussion section in the following way:

“One possible criticism of continuous psychophysics is that it introduces additional unmeasured factors such as action variability, intrinsic costs, and subjective internal models. While classical psychophysical paradigms take great care to minimize the influence of these factors by careful experimental design, they can nevertheless still be present, e.g. as serial dependencies (Green, 1964; Fischer and Whitney, 2014; Fründ et al., 2014). Similarly, estimates of perceptual uncertainty often differ between classical psychophysical tasks when compared directly. For example, Yeshurun et al., (2008) reviewed previous psychophysical studies and conducted experiments to empirically test the theoretical predictions of SDT between Yes-No and 2IFC tasks. They found a broad range of deviations between the empirically measured perceptual uncertainty and the theoretical value predicted by SDT. Similarly, there are differences between 2AFC tasks, in which two stimuli are separated spatially, and their equivalent 2IFC tasks, in which the stimuli are separated in time (Jäkel and Wichmann, 2006). While the former task engages spatial attention, the latter engages memory. These factors are typically also not accounted for in SDT-based models. Thus, substantial empirical evidence suggests that factors often not modeled by SDT, such as attention, memory, intrinsic costs, beliefs of the subject differing from the true task statistics, and learning nevertheless very often influence the behavior in psychophysical experiments.”

– The manuscript appears both to misleadingly describe findings in Bonnen et al., (2015), and to self-contradict its own assertions regarding those findings. The authors several times write that Bonnen et al.,'s tracking-based estimates of position uncertainty (i.e. position discrimination thresholds) are an order of magnitude (i.e. ~10x) higher than corresponding forced-choice-based estimates. The authors contrast this discrepancy with their own results, saying that they 'infer values that differ by a factor of 1.36 on average'. It is true that the raw estimates of Bonnen et al., reported tracking-based estimates that were ~10x larger. But that was before Bonnen et al., took several obvious factors into account (e.g. the benefit of integrating across 15 frames of each stimulus presentation, the predicted improvement in sensitivity in 'two-look' forced choice tasks). After taking these factors into account (as it is appropriate to do), the ~10x difference between the tracking- and forced-choice-based was reduced to a ~2x difference. In the supplement, the authors acknowledge that Bonnen et al., took these factors into account, but their writing in the abstract, introduction, and discussion (references to order of magnitude differences) reads as if they are unaware. I am unclear about the reasons for this discrepancy, but it should be corrected.Further, on page 5, left column the authors report that estimates of perceptual uncertainty of target position from the Kalman Filter model differ from the forced-choice-based estimates by approximately 2x. They write: ‘The average factor between the posterior mean perceptual uncertainty in the continuous task and the 2IFC task in the 5 higher contrast conditions is 1.20 for the LQG model, while it is 1.73 for the KF. Only in the lowest contrast condition, it increases to 2.20 and 2.93 arcmin, respectively.’. This difference is in line with the conclusions of Bonnen et al., and does not square with the assertion that tracking-based estimates were off by an order of magnitude (~10x). Given that the current Kalman filter analysis essentially replicates the analysis performed by Bonnen et al., these two assertions (10x differences vs 2x differences) cannot both be correct. Again, in keeping with the first point above, all this seems like a forced attempt to convince the reader that observed deviations are small between the tracking- vs. forced-choice-based estimates of perceptual uncertainty about target position. But it seems unnecessary. Model 4 does a more accurate job of estimating perceptual uncertainty about target position than simpler models (and seems to provide a closer match to forced-choice-based estimates). Let those results stand on their own.

Again, thank you for pointing out that this could potentially be misunderstood by a reader. But, we would like the reviewer to be as generous with us as with the authors of the Bonnen et al., study. We were referring to the fact that in the Bonnen et al., (2015) paper in Figure 10 the factor between forced choice and tracking is approximately one order of magnitude. It is worth pointing out that Bonnen et al., (2015) discuss that this factor can be reduced by accounting for temporal integration.

We understand that the way we have referred to these values could potentially be read in a misleading way. We agree that our results can stand on their own simply due to the fact that the estimates are closer to forced-choice based estimates. Therefore, in the revised manuscript all references to "an order of magnitude" are removed and instead we simply state the values as they result from our reanalysis of the data from Bonnen et al., (2015).

We have now written in the abstract,

“However, what has precluded wide adoption of this approach is that current analysis methods do not account for the additional variability introduced by the motor component of the task and therefore recover perceptual thresholds, which are larger compared to equivalent traditional psychophysical experiments.”

and rewritten the parts that mention “orders of magnitude” in the introduction and discussion as well.

– For a computational paper, in which the primary contribution is to develop methods for data analysis and parameter estimation, there should be substantially more discussion of which parameters have their values trade off of one another. For example, the authors have taken the trouble visualize a posterior distribution over the parameters (Figure 3A). They should help the reader develop an intuition for why the posterior has the structure that it does. All models other than the Kalman filter model include perceptual uncertainty about target position (\σ), perceptual uncertainty about cursor position (\σ_p), and motor variability. All of these quantities affect the reliability with which discrepancies between target and cursor position can be evaluated, and should have similar effects on performance. The figure shows that these parameters trade-off with each other (strong negative correlations in the posterior distributions over the model parameters). The authors should include some discussion of these trade-offs, helping to provide the reader intuition for why these parameters trade-off in the way that they do. Similarly for the positive correlation between motor variability (\σ_m) and perceptual uncertainty about cursor position.

Sorry, but we first need to clarify a conceptual point about Bayesian inference. While there are some correlations in the two-dimensional marginals of the posterior distributions, these cannot be interpreted as perfect linear trade-offs. Instead, there are non-linear relationships between multiple variables in the posterior distributions and one should always be cautious when interpreting posterior distributions using only two-dimensional marginals. The mode of the full non-Gaussian joint posterior is also not in general equal to the mode of the marginal distributions.

For example, there is a positive correlation between motor variability (parameter σ_m) and behavioral costs (parameter c) in the two-dimensional marginal posterior distribution shown in Figure 3A. The posterior distribution infers that these parameters lie in a certain range: e.g., most of the posterior mass for σ_m is between 0.48 and 0.52. Within that range, there is some remaining uncertainty about which exact values the parameters take. However, this does not mean that one can arbitrarily increase perceptual uncertainty while linearly increasing the cost, because this relationship is non-linear and breaks down outside of the region with high posterior density. Similarly, the Figure2—figure supplement1 “Pareto efficiency plot” is a clear example that tradeoffs can interact highly nonlinearly. We also want to emphasize that the ability to infer a full joint posterior distribution (with correlations between variables) is a desirable property of Bayesian statistics instead of a shortcoming. With a maximum-likelihood approach, we would obtain a single best-fitting value per parameter without any insight into its associated uncertainty.

We agree that, intuitively, it may seem like “these quantities affect the reliability with which discrepancies between target and cursor position can be evaluated, and should have similar effects on performance”. But, importantly, our model-based analysis shows that their influence is not the same. This can be seen in our parameter recovery analysis, which shows that our inference method can recover the true values of these parameters used to simulate the trajectories quite well (Figure 3B). Note that our inference method considers full trajectories in the space of states, sensory measurements, actions and associated noises. Thus, the inference is based not on summary statistics such as position errors or cross-correlations, but takes into account how the different parameters interact at each time step.

All these points together emphasize the importance of including extensive simulations to check for parameter recovery (‘Simulations results’) and proper model comparison on experimental data (‘Model comparison’).

As for a more in depth discussion of the influence of the parameters, we have decided to substantially expand the final paragraph of the section “Computational models of psychophysical tasks”. This includes a qualitative discussion of the effects of most of the parameters when introducing the different models and will hopefully aid in developing better intuitions about the behavior of the overall model and the inferred properties. Here are some example sentences:

“[…] the KF has only one free parameter, the perceptual uncertainty \σ. This parameter describes the uncertainty in the sensory observation of target's true position.”

“The difference lies in the cost function, which now can additionally accommodate internal behavioral costs c (Figure 2 C) that may penalize e.g. large actions and thus result in a larger lag between the response and the target. The control cost parameter c therefore can implement a trade-off between tracking the target and expending effort (see Figure 2—figure supplement1).“

“[…], the subject could instead assume that the target follows a random walk on position with standard deviation \σ_s and an additional random walk on the velocity with standard deviation σ_v. These subjective assumptions about the target dynamics can, for example, lead the subjective actor to overshoot the target, for which they then have to correct.”

– Regarding their analysis of simulated data, the authors report that attempts to infer the value of the target position uncertainty (\σ) from Model 4 were accurate (Figure 3C). It would be helpful for the authors to describe in more detail the parameters of the simulation. Specifically, in would be useful to explain and provide intuition for how, if the observer had a mistaken belief regarding the drift dynamics, it is possible to accurately infer the value of perceptual uncertainty about target position. A naïve reader might presume that, because there is an infinite number of pairs of position uncertainty (\σ) and presumed drift variance (\σ_s) that determine the Kalman gain, that \σ could not in fact be accurately estimated if \σ_s does not equal the true drift parameter (\σ_rw). Please explain for the reader how this works.

Sorry, but this again refers back to the issue of properties of Bayesian inference of parameters in the present models. This is why validation of inference on synthetic data and proper model comparison are fundamentally important.

For what it is worth, our intuition for why we can disentangle the subjective random walk drift from the perceptual uncertainty goes as follows: As you note correctly, both parameters influence the Kalman gain. However, the state estimate (x^hat_t) is not only a function of the Kalman gain, but also of the concrete sensory observations (y_t) that the subject receives (see equation 5). These sensory observations depend on the perceptual uncertainty, but not on the subjective random walk drift.

Another way to get an intuition is to see that the prediction of the next position may be rendered uncertain in equivalent ways by the two factors but with new incoming sensory observations, there is less variability in the estimates when the perceptual uncertainty is low. This leads to a difference in behavior that the statistical model picks up.

Additionally, to show that both the perceptual uncertainty and the subjective standard deviation of the random walk can be inferred, we ran the same simulations as for Figure 3A and B (which were previously for the bounded actor) for the subjective actor. The results show that all of its parameters can be inferred accurately. The updated versions of Figure 3A and B now show all six parameters of the subjective actor model.

ToneThe current paper describes shortcomings of the Bonnen et al., (2015) data analyses. Many of these shortcomings were explicitly acknowledged by Bonnen et al., It would help the tone and tenor of the manuscript if, when the current authors describe the acknowledged shortcomings, the current authors cite Bonnen et al., (2015).Examples include:– "A model, which only considers the perceptual side of the task, will therefore tend to overestimate perceptual uncertainty because these additional factors get lumped into perceptual model parameters, as we will show in simulations."– "A model without these factors needs to attribute all the experimentally measured behavioral biases and variability to perceptual factors, even when they are potentially caused by additional cognitive and motor processes."The current paper makes a nice contribution in demonstrating these points quantitatively. But, in places (and in keeping with the flavor of remarks above), the writing seems concerned that readers might not recognize the paper's unique contribution. I think the paper's contribution is clear. And I think that their paper will read better, and leave a better impression, if it looks for ways to portray previous work in a more generous light.

We are sorry that we have written some passages in a way that the reviewer feels potentially could be understood as a misrepresentation of Bonnen et al., (2015). This was not our intent. The other reviewers did not comment on this. In the passages cited above, we do not refer to the data analysis by Bonnen et al., (2015), specifically. For example, the statement that “a model, which only considers the perceptual side of the task, will tend to overestimate perceptual uncertainty” seems fairly neutral to us. We are making a general statement that a model without a motor component has to attribute all the variability and biases in the tracking data to perception. This is simply a fact that is clearly shown in our simulations in Figure 3C.

However, there is still a difference between mentioning factors such as actions and motor variability and explicitly accounting for these factors in a quantitative computational model, whose implementation is provided to the research community.

Nevertheless, to acknowledge the comments of the reviewer, we have revised a paragraph in the introduction and now explicitly mention that Bonnen et al., (2015) acknowledge that motor processes play a role in the tracking task. If there are any other specific passages that paint the contributions of Bonnen et al., (2015) in an unfavorable light, we are more than happy to consider changing them.

The revised paragraph reads:

“The KF models the perceptual side of the tracking task only, i.e. how an ideal observer sequentially computes an estimate of the target’s position. Tracking, however, is not merely a perceptual task but involves motor processes as well: In addition to the problem of estimating the current position of the target, a tracking task encompasses the motor control problem of moving the finger, computer mouse, or gaze towards the target. This introduces additional sources of variability and bias. First, repeated movements towards a target exhibit variability (Faisal et al., 2008), which arises because of neural variability during execution of movements (Jones et al., 2002) or their preparation (Churchland et al., 2006). Second, a subject might trade off the instructed behavioral goal of the tracking experiment

With subjective costs, such as biomechanical energy expenditure (Di Prampero,1981) or mental effort (Shenhav et al., 2017). Third, subjects might have mistaken assumptions about the statistics of the task (Petzschner and Glasauer, 2011, Beck et al., 2012), which can lead to different behavior from a model which perfectly knows the task structure, i.e. ideal observers. Bonnen et al., (2015) acknowledge that motor processes play a role, but their computational model, the KF, does not account for these processes. A model that only considers the perceptual side of the task will, therefore, tend to overestimate perceptual uncertainty because these additional factors get lumped into perceptual model parameters, as we will show in simulations.”

Specific Comments– Experimentor/Participant distinctionThe experimenter has access to the true target position and the true cursor position is available. The experimental participant him/herself has access only to the estimates of target and cursor positions. The article should more explicitly discuss this issue in the main text. Figure 2 and the Supplemental derivations allude to it, but it should be discussed more prominently.

We discuss this more prominently in the revised manuscript first in the introduction:

“By modeling the particular task as it is implemented by the researcher allows deriving a normative model of behavior such as ideal observers in perceptual science (Green and Swets,1966; Geisler, 1989) or optimal feedback control models in motor control (Wolpert and Ghahramani, 2000; Todorov and Jordan, 2002). This classic task analysis at the computational level (Marr, 1982) can now be used to produce predictions of behavior, which can be compared to the actual empirically observed behavior. However, the fundamental assumption in such a setting is that the subject is carrying out the instructed task and that subject's internal model of the task is identical to the underlying generative model of the task employed by the researcher. Here, instead, we allow for the possibility that the subject is not acting on the basis of the model the researcher has implemented. Instead, we allow for the possibility that subjects' cost function does not only capture the instructed task goals but also the experienced subjective cost such as physiological and cognitive costs of performing actions. Such an approach is particularly useful in more naturalistic task setting, where subjective costs and benefits of behavior are difficult to come by a priori.

Similarly, we allow subjects' subjective internal model of stimulus dynamics to differ from the true model employed by the researcher in the specific experimental task.

In the spirit of rational analysis (Simon, 1955; Anderson, 1991; Gershman et al., 2015) we subsequently invert this model of human behavior, by developing a method for performing Bayesian inference of parameters describing the subject. Importantly, inversion of the model allows all parameters to be inferred from behavioral data and does not presuppose their value, so that, e.g., if subjects' actions were not influenced by subjective internal behavioral costs, the internal cost parameter would be estimated to be zero. This approach therefore reconciles normative and descriptive approaches to sensorimotor behavior.”

and explain the graphical models in Figure 1 in more detail. Specifically, we have expanded the beginning of the section “Bayesian inverse optimal control” in the following way to better explain the distinction between the researcher’s and the subject’s perspective:

“The normative models described in the previous section treat the problem from the subject's point of view. That is, they describe how optimal actors with different internal models and goals should behave in a continuous psychophysics task. From the subject's perspective, the true state of the experiment x_t is only indirectly observed via the uncertain sensory information y_t (see Figure 1D). From the point of view of a researcher, the true state of the experiment is observed because they control the experiment, e.g. using a computer that presents the target and mouse position. The goal is to estimate parameters \theta that describe the perceptual, cognitive, and motor processes of the subject for each model, given observed trajectories x_{1:T} (Figure 1F).“

and the Discussion section also includes

“On a conceptual level, the results of this study underscore the fact that psychophysical methods together with their analysis methods always imply a generative model of behavior (Green and Swets, 1966; Swets, 1986; Wixted, 2020).

Nevertheless, different models can be used in the analysis i.e. models that may or may not be aligned with the actual generative model of the experiment as designed by the researcher or with the internal model that the subject is implicitly using. For example, while classical SDT assumes independence between trials and an experiment may be designed in that way, the subject may assume temporal dependencies between trials.

The analysis framework we use here accounts for both these possibilities. A participant may engage in an experiment with unknown subjective costs and false beliefs, as specified in the subjective actor model. Similarly, this analysis framework also allows for the researcher to consider multiple alternative models of behavior and to quantify both the uncertainty over individual models' parameters as well as uncertainty over models through Bayesian model comparison.”

– Explanation of matrix values.The authors should explain why the B=[0 dt]' parameter takes on the particular values that it does. Is the implication that the control action is best understood as a 'rate of change' so that result matrix-multiplying the control action with B is a position? Please explain. The value of the non-zero element of B trades off perfectly with the square root (?or similar?) of the movement cost parameter 'c'. So it would be valuable for the authors to explain why they chose to set B to have the value it does.

The non-zero element of B is the gain applied to the action to yield the change in position. As you explain correctly, the choice of the non-zero element of B trades off with the action cost parameter c. If we had set it to a different value, we would simply have a different scaling of the action cost parameter. This is why B is not a free parameter of the model but set to a fixed value. Setting this value to the duration of a time step is customary in modeling of dynamical systems including in motor control, because it allows interpreting the magnitude of the action in physical terms, i.e. as a velocity.

Telegraphic mathematical development.In the supplement, the mathematics associated with transitioning between Supplemental Equations S16-S18 needs to be expanded. Conditioning the nD Gaussian on x_t and marginalizing out x_hat_t are two separate steps, but the manner in which the passage is written encourages the reader to presume that they are a single step. Too much analytical work is foisted on the reader if he/she wants to carefully follow the derivation along. The derivation is correct (I worked it out myself), but it should be laid out more explicitly. Further, the final expression for the likelihood of the state (x_{t+1}) on time step t+1 should be provided. This likelihood is straightforwardly obtained by marginalizing out the estimate (x^{hat}_{t+1}) of the (2-vector) state, but it would be nice for the reader if the final expression was actually made explicit for the reader.

Thanks for verifying our derivations! We have inserted some details between Equations S16 and S18 (which is now Equation S20) to hopefully make the derivations easier to follow. Specifically, we have treated the marginalization and conditioning as separate operations. We have also clarified how x^{hat}_{t+1} can be marginalized out from p(x_{t+1}, x^{hat}_{t+1} | x_{1:t}).

Subjective model of state dynamicsThe authors should make explicit in equation form the fallacious state dynamics that are assumed by the subjective observer. The authors describe it in words, but equations will prevent any uncertainty that the description may produce. Something like:position walk: p_{t+1} = p_t + eps; where eps ~ N(0, σ_{rw})velocity walk: v_{t+1} = v_t + eta; where eta ~ N(0, σ_v )x_{t+1} = p_{t+1} + sum( v_{ 0:(t+1) } ) where the sum goes from 0 to t+1

Thank you for this suggestion. We have added the univariate descriptions of the subjective position and velocity random walks to the supplementary text to clarify the subjective dynamics. We have also extended Appendix 2 Table 1 to clarify the dynamics of all models.

Subjective model of estimate of velocityAlso, it is not made completely clear how the subjective observer's fallacious estimate of velocity is computed, how it resides in the model, whether the state is expressed as a 2-vector [ x_t x_p ] or a 3-vector [ x_t x_p v_t ], and whether the estimate of the state is expressed as a two-vector [ xhat_t xhat_p ] or a 3-vector [ xhat_t xhat_p vhat_t ]. An expanded discussion of these issues should be included. The expressions are correct but as they stand, but the mathematical development and discussion of these points are a bit too telegraphic.

Sorry, the definition of the state space for our models could indeed have been clearer. We have extended Appendix 2 Table 1 to include definitions of the state vectors and added a statement in the main text pointing these differences out.

Typo in the supplementary equation.I believe that there is a typo in equation S2 of the supplement where the Ricatti equation is laid out. The term with the inverse currently reads ( C*P*C' )^-1 whereas this term in standard expressions for the Ricatti equation should be ( C*P*C' + W*W' )^-1. This term follows the same form as the inverse term in the expression for the Kalman gain (see the line above) which reflects the total covariance ( prior + observation covariances ).

Thanks for catching this error, we fixed it. The error was only in the equation and does not affect the implementation.

Reviewer #3 (Recommendations for the authors):The paper is already in a very polished state. I have only a few comments about clarity and completeness.– I found myself getting confused while reading the paper about what parameters were actually involved in each model (e.g. pg 3: "the KF model has only one parameter: perceptual uncertainty σ"). My recommendation to the authors is to move equations 1-4 out of the Methods section and into the main text. Personally I would consider this part of the results (ie. "what are the models we're using"?) and so I would rather see this presented pedagogically within the main paper. Keep implementation details or other technical issues in the Methods, but present the models themselves in the Results section. Just a recommendation, but that's my 2 cents!

This is a good idea! We have moved these equations into the main text. We have also expanded the verbal description of the models in the section “Computational models of psychophysical tasks” and included a reference to Appendix 2 Table 1, which clarifies what parameters are included in which model. In addition, we have changed some of the model descriptions in the section “Computational models of psychophysical tasks” to hopefully improve the comprehensibility of this section.

– This leads to a related note on clarity: it's not entirely clear to me which parameters are being fit in each model. Presumably you're not fitting the C matrix in equation 2? What about B in equation 1? Come to think of it, I'm not sure where the cursor fits in, allowing for uncertainty in cursor position). (Does that become part of the state vector x_t? Please unpack this more clearly so that we can understand what these equations correspond to for each of the models discussed in the paper!

We hope that our changes in the section “Computational models of psychophysical tasks” as mentioned in our response to the previous comment clarify this!

– "Our implementation will be made available on github" -- I want to note that this should be a requirement for acceptance! (i.e., the code should be posted before the paper is accepted).

Of course. The implementation is available at https://github.com/RothkopfLab/lqg and we have added the link in the manuscript.

– The name "bounded actor" seems like a poor one, since there aren't any bounds on the actor. (There are just costs). Figure 2 C refers to it as "bounded actor with behavioral costs" -- personally I would keep the "with behavioral costs" and drop the "bounded".

We agree that, formally, there are no bounds as this term is used within the optimization literature. Instead, we use this term as it is customary in parts of the cognitive science literature, where, according to Herbert Simon’s definition, bounded rationality takes the limitations on cognitive capacity into account. In our model, the costs capture both biomechanical costs in the motor system and cognitive costs.

– One other point that would be worth making: you could also formulate a "with subjective belief" version of any of the simpler models. Even in the simple KF model, you could allow for mismatch between the true dynamics and the subject's belief about the dynamics. I think it's probably ok to leave the model comparisons as is, since obviously it becomes a bit messy if we have to include "optimal" and "subjective" versions of each of the models, but you should at least mention somewhere in the paper that this can be used to improve the accuracy (in terms of matching observer behavior) for any of the models.

Yes, we fully agree that we could compare any possible combination of models with or without subjective beliefs, action variability, and behavioral costs. We decided against this for two reasons.

First, the introduction of behavioral costs leads to the largest difference in the model comparison (Figure 4E), while the subjective beliefs make a smaller difference. Thus, it would be unlikely that a KF with subjective beliefs would fare better than the other models.

Second, and more importantly, the subjective actor is a generalization of the other models, so if action variability and costs did not play a large role but subjective beliefs were important, our method would infer posterior distributions over the action variability and cost parameters that are essentially zero but the posterior distributions over the subjective beliefs would be very different from the true dynamics, i.e. we would essentially recover a KF with subjective beliefs. This is not what we find, though (Figure 4B and C).

Following your suggestion, we point the following out in the section introducing the subjective actor:

“One could also formulate a version of any of the other models to include subjective beliefs. However, the subjective actor model is a generalization that includes all of these possible models. Accordingly, if the subjective beliefs played an important role but the costs or variability did not, we would infer the relevant parameters to have a value close to zero.”

– Can you say anything about the discrete time assumptions of the model? i.e., how would you expect the model parameters (or the accuracy of the fit) to change if you switched to 120 Hz frame rate or 20 Hz frame rate?

For the analyses in the present paper, we have chosen the sampling rate to match the frame rate at which the experiment was conducted. If one were to change the frame rate, the statistics of the target’s random walk would no longer be captured by the model in a straightforward way. For example, if we fit the model at a frame rate of 120Hz to data that was collected at 60Hz, the assumption of linear dynamics with independent Gaussian noise would no longer hold, because there would only be noise at every second time step.

Theoretically, if one were to run a tracking experiment at a different frame rate, we would certainly expect a higher uncertainty in the parameter estimates at lower frame rates because there would be less data. At higher frame rates, on the other hand, one could run into problems with the assumption of independent noise across time steps, as discussed in our response to Reviewer #1.

– Finally, a small note on grammar/usage: the paper frequently uses "… , which…" in cases where "that" (with no comma). e.g., in the abstract: "recover perceptual thresholds, which are one order of magnitude larger compared to equivalent traditional psychophysical experiments". This makes it sound like you're providing a definition of threshold, rather than describing a property of the recovered thresholds. More correct would be: "We recover perceptual thresholds that are one order of magnitude larger than …". I found this issue repeatedly in the text. You could basically look for nearly every occurrence of ", which" and replace by "that".

Thanks for pointing this out. We have changed several of our "which" constructions to "that".